

**ShellTrace v1.0 - A new approach for modelling growth and trace element uptake in marine**
**bivalve shells: Model verification on pacific oyster shells (*Crassostrea gigas*)**
*de Winter, Niels J.*
Analytical, Environmental and Geochemistry (AMGC) Department, Vrije Universiteit Brussel, Pleinlaan 2, 1050
Brussels, Belgium



**Abstract**
Bivalve shells record changes in their environment in the chemical composition of their shells and are
frequently used as paleoclimate archives. However, many studies have shown that physiological changes
related to growth of the bivalve may overprint these chemical tracers. In the present study, a new approach is
presented that models growth and development of bivalve shells without a priori knowledge of the physiology
of the species. The model uses digitized growth increments in a cross section of the shell together with basic
assumptions of the shape of the shell in order to model changes in shell length, thickness, volume, mass and
growth rate at a daily resolution through the lifetime of the bivalve. This approach reconstructs the growth
history of bivalves based on their shell without the need for a culture experiment, paving the way for growth
rate estimations based on fossil bivalve shells. Combination of the growth model with 2D X-Ray Fluorescence
trace element mapping allows the incorporation of trace elements into the shell to be modelled in 3D through
the bivalve's lifetime. This approach yields records of integrated total-shell trace element concentrations and
accumulation rates, which shed light on the rates and mechanisms by which these trace elements are
incorporated into the shells of bivalves. Application of growth and trace element modelling on a set of modern
pacific oyster shells of well-known origin and comparison of model results with conventional trace element
transects highlights the importance of considering heterogeneity in mineralogy, mineralization rates and
chemical composition within the shells of bivalves. These insights lead to a better understanding of the
complexity of trace element concentrations in bivalve shells, which can then be applied as proxies for the
reconstruction of sub-annual changes in palaeoenvironmental conditions over geological timescales.



## 1. Introduction

The study of climate and environmental change over geological timescales has yielded various important
insights into the dynamics of climate systems on Earth (e.g. Huber et al., 1995; Hesselbo et al., 2000; Zachos et
al., 2001; Wang et al., 2001; Sluijs et al., 2006). The knowledge about the sensitivity of Earth's climate and
environment that results from these studies is indispensable for the prediction of future changes in Earth's
climate. The study of environmental changes relies both on proxy-based palaeoenvironmental reconstructions
(e.g. McDermott, 2004; Leng and Marshall, 2004; Zachos et al., 2006; Affek et al., 2008) and climate and
environmental modelling based on, and calibrated with, these reconstructions (e.g. Barron et al., 1984;
Kutzbach et al., 1989; Claussen et al., 2002; Andrews et al., 2012). Important archives for proxy-based
reconstructions of palaeoenvironment on a high temporal resolution are fast-growing carbonate records such
as speleothems and the fossil skeletal remains of calcifying organisms such as corals, brachiopods and molluscs
(Goreau, 1977; de Winter and Claeys, 2017; Ullmann et al., 2017; Vansteenberge et al., CHEMGEO; de Winter
et al., PPP). Mollusc shells have gained much attention in the last decades, because the calcite in these shells
has high fossilization potential, their populations are abundant and several studies have shown that chemical
proxies in these shells record changes in their environment (e.g. Klein et al., 1997; Schöne et al., 2003; Lazareth
et al., 2003; Gillikin et al., 2008). Stable isotope ratios of carbon and oxygen in the calcite shells of bivalve
molluscs are almost exclusively precipitated in equilibrium with the surrounding seawater and can thus be
used to trace changes in temperature, productivity and salinity on a seasonal scale (Klein et al., 1996; Kirby et
al., 1998; Goodwin et al., 2001; Ullmann et al., 2010). However, to disentangle the effects of these parameters
and to properly understand changes in the local environment, it is important to apply multi-proxy studies of
shell calcite (e.g. Takesue and van Geen, 2004; Ullmann et al., 2013; de Winter et al., PPP). It is for this reason
that bivalve sclerochronology studies have focused on understanding the relationships of trace element
concentrations in bivalve calcite with their environment (Lorrain et al., 2005; Wanamaker et al., 2008; Freitas
et al., 2009; Schöne et al., 2011). Since then, a range of trace element ratios (e.g. Mg/Ca, Sr/Ca, Ba/Ca, Mn/Ca
and Li/Mg) have been used as proxies for environmental parameters (Klein et al., 1996a; Lazareth et al., 2003;
Carré et al., 2006; Gillikin et al., 2008; Füllenbach et al., 2015; Vihtakari et al., 2017).
A few studies have focused on the development and chemical composition of modern oyster shells and its
relation to the environment (e.g. Palmer and Carriker, 1979; Carriker et al., 1980; Lee et al., 2008; Surge and
Lohmann, 2008; Ullmann et al., 2010; 2013). These studies have shown that oyster shells are composed mostly
of calcite occurring as foliated calcite layers, prismatic calcite and chalky calcite while the myostracum and
hinge ligament are made of aragonite (Stenzel, 1963; Palmer and Carriker, 1979). There is some discussion
about the role of these calcite mineral phases, whether their precipitation is controlled by environmental
conditions and whether changes in the precipitated mineral phase are paced to regular (solar or lunar) cycles
(Carriker et al., 1980; Kirby et al., 1998; Surge et al., 2001; Ullmann et al., 2010). It has even been proposed
that the mineralization of the chalky calcite phase in oyster shells is mediated by microbial activity (Vermeij,
2014). Beside mineralogy and chemistry of the shell, shell growth rate and dimensions vary widely between



individuals in response to several environmental factors such as growth space, substrate, food availability and
amount of predation (Galtsoff, 1964; Palmer and Carriker, 1979; Surge and Lohmann, 2008).
All these physiological changes, such as variations in growth and metabolic rate, shell mineralogy and
spawning events, which affect the incorporation of trace elements into the shell of bivalves, complicate the
use of trace element records to complement environmental reconstruction by stable isotope
sclerochronology, (Klein et al., 1996b; Gillikin et al., 2005; Immenhauser et al., 2005; Freitas et al., 2006).
Furthermore, several studies have shown that rates by which trace elements are incorporated into bivalve
shells and the degree to which these rates are controlled by environmental factors can be vastly different
between different bivalve species (Reinfelder et al., 1997; Steuber, 1999; Richardson et al., 2004; Carré et al.,
2006). To constrain such variations in physiological parameters on the chemistry of bivalve shells, species-
specific culture experiments are carried out under controlled circumstances so relationships between
environmental parameters and shell chemistry can be precisely determined (e.g. Wang and Fisher, 1996;
Freitas et al., 2006; Gillikin et al., 2006). Such experiments can only be executed on extant bivalves, which
severely limits the potential to apply the acquired proxy transfer functions to reconstruct climate and
environment in pre-Cenozoic times (e.g. de Winter et al., PPP). In this study, a model is introduced that
approximates the development of a range of size parameters in the bivalve shell through ontogenetic age,
based solely on digitized coordinates of recognized annual shell increments in a longitudinal cross section
through the shell. Additionally, the modelled growth development and recruitment pattern in the shell cross
section is then superimposed on an XRF trace element map to model the incorporation of trace elements into
the shell with age. The application of this growth and trace element model is demonstrated on a set of shells
of the modern pacific oyster (*Crassostrea gigas*) with well-known origins and dimensions. Model results are
compared with conventional trace element analyses on line scans through the hinge of the shells as well as
with results from previous bivalve growth studies.
**2. Materials and Methods**
**2.1 Specimen acquisition and preparation**
A set of eight modern pacific oyster (*Crassostrea gigas*) shells were obtained from restaurant Jardin van Gogh
in Brussels, Belgium (http://www.jardinvangogh.be). The oysters originate from a cultivation area in coastal
Normandy (France, 49°4.0' N latitude and 1°35.47' W longitude) and were harvested on February 13[th] 2017.
The shells were rinsed with acetone ($C_3H_5OH$) and distilled water, cleaned superficially with a soft brush and in
an ultrasonic bath and oven dried overnight at 50°C. Dried shells were weighed on a digital scales ($\sigma = 0.01$ g),
their dimensions (shell length, maximum shell width, maximum shell thickness) were measured using digital
callipers ($\sigma = 0.01$ mm) and their volume was determined by water displacement measurement using a graded
cylinder. All shells were embedded in Araldite® 2020 epoxy resin (Huntsman, Basel, Switzerland), sectioned
longitudinally along their axis of maximum growth using a slow rotating, diamond coated saw (Ø = 1 mm) and
high-grade polished using silicon carbide polishing disks (up to P2400 grain size). Polished shell surfaces were





imaged by colour scanning (RGB) using an Epson® 1850 flatbed scanner (Seiko Epson Corp., Nagano, Japan) at
a pixel resolution of 6400 dpi (± 4 µm pixel size).
**2.2 X-Ray Fluorescence measurements**
Concentrations of calcium (Ca), silicon (Si), magnesium (Mg), strontium (Sr), zinc (Zn), sulphur (S), phosphorous
(P), manganese (Mn) and iron (Fe) were measured on the polished shell surfaces using a Bruker® M4 Tornado
micro-X-ray Fluorescence scanner (Bruker GmbH, Berlin, Germany) equipped with a Rh X-Ray source using
maximum energy settings (50 kV, 600 µA) with a spot size of 25 µm (Mo Kα) and two Silicon Drift detectors.
The XRF setup is described in detail in de Winter and Claeys (2017). The entire shell surface was mapped semi-
quantitatively by XRF scanning in mapping mode using 1 ms integration time per pixel (as described in de
Winter and Claeys, 2017). Spacing between pixels was chosen in such a way that the total amount of pixels per
map was relatively constant (±4.0*10$^6$) for all shells while choosing the minimum rectangular area that
contained the entire cross section area. This caused pixel spacing in maps to vary between 25 µm (interlocking
X-Ray spots) and 30 µm. Quantitative XRF line scans were carried out on the dense foliated calcite layers in the
hinges of all shells perpendicular to the growth layers (see Palmer and Carriker, 1979) using the point-by-point
scanning method outlined in de Winter et al. (in review, PPP) with an integration time of 60 seconds per point.
This integration time allowed enough count statistics for the instrument to reach the Time of Stable
Reproducibility (TSR) and provide reproducible trace element concentrations for the elements of interest (de
Winter et al., 2017). All XRF line scans were quantified using the Bruker Esprit® fundamental parameters (FP)
quantification relative to the BAS CRM 393 limestone standard. Errors of reproducibility of µXRF
measurements are generally higher than the instrumental error and depend on the integration time and the
excitation energy of the element (see de Winter and Claeys, 2017; de Winter et al., 2017). Typical
reproducibility errors of µXRF point measurements are reported in **Table 3**.
**2.3 Data preparation**
Concentrations (in µg/g) of trace elements were calculated for profiles measured using XRF and plotted using
Grapher™ 8 (Golden Software Inc., Golden, CO, USA) graphing software. Timing of shell deposition was
inferred from annual cyclicity in trace element profiles. Growth increments (lines of simultaneous deposition in
the shell) were digitized on high-resolution colour scans of polished shell cross sections of the shells using the
pen tool in Adobe Illustrator® CC 17.1.0 (Adobe Systems Inc., San Jose, USA). Outlines of the rectangular area
of the cross section mapped by XRF were digitized in the same way. Line coordinates were saved in a SVG-file,
which allowed X-and Y-coordinates of the lines to be extracted and ordered into a comma separated (CSV) file
to be imported into the modelling script (**Step 1** in **Fig. 1**). An example of a shell cross section with traced
growth increments is shown in **Figure 2**. SVG- and CSV files of growth increments digitized in all shells used in
this study are found in **supplementary data 1**.
XRF map data was processed using Bruker Esprit® software. Maps were subject to a PCA-assisted maximum
likelihood phase analysis using a selection of distinctive elements (Ca, Mg, Sr, P, S and Mn). Minimum phase



area was fixed to 0.05% of the total map area. Phase analysis results were matched with interpreted growth
increments and high-resolution colour scans and associated phases were merged. Sum XRF spectra of all pixels
in each phase were quantified relative to the BAS CRM 393 standard using Esprit® software. Minimum area of
each phase was such that the total integration time contained in all the pixels allowed Time of Stable
Reproducibility to be reached for the quantification of the sum spectrum (de Winter et al., 2017). Phase maps
were exported as BMP files and oriented in the same way as colour scans with the shell hinge facing left and
the inside of the shell facing down (see **Fig. 2**).
**2.4 Modelling approach**
A modelling routine was written in the open source computational software package R (R Core Team, 2013)
using Microsoft® Visual Studio Code Version 1.10.2. Shell growth and trace element accumulation rates were
modelled in six steps, of which **Step 1** is a data preparation step (see above), **Step 2-4** form the growth model
and **Step 5-6** make up the trace element model (**Fig. 1**). The complete R-script used for the model is provided
in **supplementary data 2**. Variables used in the modelling process are indicated in **Figure 3**.

**2.4.1 Growth modelling**
**Step 2** of the model converts X- and Y-coordinates of all digitized increments to millimetres using the ratio
between the real length of the digitized image and the length in pixels. All increments are converted to one
cross section matrix (*Digitized cross section*) with a common X-axis with a default step size (*dx*) using linear
interpolation between line segments. The resulting cross section is plotted to provide a check on the model
progress. From this matrix, the area between each increment and its predecessor is calculated using the
formula:

$$\textbf{F1}: O_i = \int_{x_0}^{x_{end}} Y_{i-1}(x) - Y_i(x)\, dx \qquad (1)$$

in which $O_i$ is the area between increment *i* and increment *i*-1, $x_0$ and $x_{end}$ are the extreme values of the range
of X coordinates in *Digitized cross section* and $Y_i$ and $Y_{i-1}$ are the Y-coordinates of increment *i* and *i*-1
respectively. $Y_{i-1}$ is always bigger than $Y_i$ since bivalves build their shell by adding material on the inside of the
shell, which faces down in this model. The average shell thickness at each increment is determined using the
formula:

$$\textbf{F2}: T_i = \frac{\sum_{x_0}^{x_{end}} Y_0(x) - Y_i(x)}{x_{end} - x_0} \qquad (2)$$

in which $T_i$ is the average thickness of the shell at increment *i*, $Y_i(x)$ and $Y_0(x)$ are the Y-coordinates of
increment *i* and the top of the shell (increment 0). Total shell length is calculated from the X- and Y-
coordinates of the start- and endpoints of the increment (where the increment meets the top or bottom of the
shell) and the Pythagorean Theorem following the formula:

$$\textbf{F3}: L_i = \sqrt{(x_e - x_s)^2 + (Y_e - Y_s)^2} \qquad (3)$$



in which $L_i$ is the length of the shell at increment i, $x_s$, $x_e$, $Y_s$ and $Y_e$ are the X- and Y-coordinates of the start- and
endpoints of the increment *i*. The results of these calculations, as well as values for $x_s$, $x_e$, $Y_s$ and $Y_e$ are stored
in a *Matrix of parameters by increment*.
**Step 3** of the growth model takes *Digitized cross section*, *Matrix of parameters by increment* and a
customizable number of increments (*N*) to be interpolated to create a new cross section matrix with *N-1*
interpolated sub-increments between each set of increments (*Sub-incremental cross section*). Interpolation of
sub-increments is done by linear interpolation of the Y-coordinate of sub-increments between that of the two
increments (see insert in **Figure 3A**) according to the following formula:
$$\textbf{F4.1}: \left[\left[Y_t(x) = Y_{i-1}(x) - \frac{n}{N} * \left(Y_{i-1}(x) - Y_i(x)\right)\right]_{n=0}^{n=N-1}\right]_{x=x_0}^{x=x_{end}} \quad (4)$$
With
$$\textbf{F4.2}: t = i - 1 + \frac{n}{N} \quad (5)$$
in which $Y_t(x)$ is the Y-coordinate of the $n^{th}$ sub-increment between increment i and increment i-1 and $x_0$ and
$x_{end}$ are the extreme values of range of X-values in *Digitized cross section* (as in **F1**). All calculated values for
$Y_t(x)$ are stored with reference to their sub-increment number (*t*) and X-coordinate in the new *Sub-incremental*
*cross section* matrix. The resulting cross section is plotted to provide a check on the model progress. This new
matrix is then used to calculate area between sub-increments, shell thickness and total shell length during
deposition of each sub-increment by formula **F1**, **F2** and **F3** respectively. Additionally, using the measured
maximum length (*Shell length*) and width (*Shell width*) of the oyster, parameters *a* and *b* of the ellipse that
forms the base of the shell for volume calculations (**Figure 3B**) are calculated according to formulae:
$$\textbf{F5.1}: a_t = \frac{1}{2} * \frac{W_{max}}{L_{max}} * (x_e - x_s) \quad (6)$$
$$\textbf{F5.2}: b_t = \frac{1}{2} * (x_e - x_s) \quad (7)$$
in which $a_t$ and $b_t$ are the parameters *a* and *b* of the ellipse that forms the base of the shell at sub-increment *t*,
$L_{max}$ and $W_{max}$ are the maximum length and width of the oyster shell and $x_s$ and $x_e$ are the X-coordinates of the
start- and endpoints of the increment *t* (as in **F3**). All these parameters are stored in *Matrix of parameters by*
*sub-increment* (**Figure 1**).
**Step 4** takes *Incremental cross section* and the ellipse parameters in the *Matrix of parameters by sub-*
*increment* to calculate the Z-values of the ellipse that forms the base of the shell at each sub-increment (see
**Figure 3B**). The Z-value is defined as the distance between the edge of the ellipse and the X-axis through the
shell (**Figure 3B**), and is calculated by the following formula, which is an adaptation of the standard formula for
ellipsoids:
$$\textbf{F6.1}: \left(\frac{Z_t(x)}{a_t}\right)^2 + \left(\frac{x^*_t}{b_t}\right)^2 = 1 \rightarrow$$





$\quad$ **F6.2**: $Z_t(x) = \left(\frac{a_t}{b_t}\right) * \sqrt{b_t^2 - x^*_t{}^2}$ $\qquad$ (8)
$\quad$ in which $Z_t$ is the Z-value (distance from X-axis) of the ellipse at X-coordinate $x^*_t$ for sub-increment $t$, $a_t$ and $b_t$
$\quad$ are the parameters of the ellipse at sub-increment $t$ and $x^*_t$ is the X-coordinate relative to the centre of the
$\quad$ ellipse, and is calculated by
$\quad$ **F6.3** $x^*_t = x - x_s - b_t$ $\qquad$ (9)
$\quad$ All Z-values are saved in a matrix (*Z-values* in **Figure 1**) with reference to their increment numbers (*t*) and X-
$\quad$ coordinates. Then, using the Z-values and the parameters from *Matrix of parameters by sub-increment*, shell
$\quad$ volume is calculated for each sub-increment. This is done by calculating areas between the sub-increment and
$\quad$ the top of the shell (sub-increment 0) in a cross sections through the shell perpendicular to the X-axis (in YZ-
$\quad$ plane, see **Figure 3C**) and multiplying these with the step size in X-direction (*dx*). This is done for every X-value,
$\quad$ and adding up all volume increments yields an estimate the total volume between the shell sub-increment and
$\quad$ the base of the shell:
$\quad$ **F7.1**: $V_t = \int_{x_0}^{x_{end}} \left(A_0(x) - A_t(x)\right) dx$ $\qquad$ (10)
$\quad$ in which $V_t$ is the volume of the shell at increment $t$ and $A_0(x)$ and $A_t(x)$ are the area under increment $t$ and
$\quad$ the top of the shell (increment 0) respectively in the cross section in YZ-direction (**Figure 3C**). These areas are
$\quad$ modelled for every X-value by constructing a circle section through the corresponding point on the sub-
$\quad$ increment in the XY cross section (centre of the YZ-cross section **Figure 3A**, or point **P1**)
$\quad$ **P1**[x, $y_1$, $z_1$] = [x, $Y_t(x)$, 0] $\qquad$ (11)
$\quad$ and the two points where the YZ-cross section intersects the ellipse that forms the base of the shell (see **Figure**
$\quad$ **3A** and **Figure 3C**):
$\quad$ **P2**[x, $y_2$, $z_2$] = [x, $Y_{ellipse}(x)$, -$Z_t(x)$] $\qquad$ (12)
$\quad$ **P3**[x, $y_3$, $z_3$] = [x, $Y_{ellipse}(x)$, $Z_t(x)$] $\qquad$ (13)
$\quad$ The value $Y_{ellipse}(x)$ is the Y-value of the ellipse with respect to the line y=0 (**Figure 3C**), which can be calculated
$\quad$ by linear interpolation of the slope of the ellipse using the start and end points of the sub-increment ($x_s$, $x_e$, $Y_s$
$\quad$ and $Y_e$) and *x*:
$\quad$ **F7.2**: $Y_{ellipse} = Y_s + \left(\frac{Y_e - Y_s}{x_e - x_s}\right) * (x - x_s)$ $\qquad$ (14)
$\quad$ The centre of this circle is the point
$\quad$ $P_c$[x, $y_c$, $z_c$] = [x, $Y_c$, 0] $\qquad$ (15)
$\quad$ its radius *r* is equal to the difference between $Y_t(x)$ and $Y_c$, and the circle can be described by the formulae:
$\quad$ **F7.3**: $\Delta y^2 + \Delta z^2 = r^2 \rightarrow$




**F7.4**: $(y_1 - y_c)^2 + (z_1 - z_c)^2 = (y_3 - y_c)^2 + (z_3 - z_c)^2 \rightarrow$
**F7.5**: $\left(Y_t(x) - Y_c(x)\right)^2 + (0 - 0)^2 = \left(Y_{ellipse}(x) - Y_c(x)\right)^2 + \left(Z_t(x) - 0\right)^2 \rightarrow$
**F7.6**: $Y_c(x) = \dfrac{(Y_t(x)^2 - Y_{ellipse}(x)^2 - Z_t(x)^2)}{2 * \left(Y_t(x) - Y_{ellipse}(x)\right)}$     (16)
**F7.7**: $r_t(x) = \sqrt{Z_t(x)^2 + \left(Y_{ellipse}(x) - Y_c(x)\right)^2}$     (17)
With all parameters known, the area in the YZ-cross section under the sub-increment (between the circle
segment and the line y=0) can be calculated as the area of the section of the circle above the ellipse plus the
area of the rectangle between the ellipse and y=0 (**Figure 3C**). The angle θ describing this circle section is equal
to:
**F7.8**: $\theta = 2 * \sin^{-1}\left(\dfrac{Z_t(x)}{r_t(x)}\right)$     (18)
However, if point **P1** lies below the ellipse ($Y_t(x) < Y_{ellipse}(x)$; in the case of irregular shells that curve upwards
during growth), the centre of the circle lies above the shell and the area under the sub-increment is described
by subtracting the section of the circle above the ellipse from the area of the rectangle (see **Figure 3C**):
**F7.9**: $A_t = \begin{cases} A_{segment} + A_{rectangle} = \frac{1}{2}\left(r_t(x)\right)^2 * (\theta - \sin\theta) + 2 * Z_t(x) * Y_{ellipse}(x), \ Y_t(x) \geq Y_{ellipse}(x) \\ A_{rectangle} - A_{segment} = 2 * Z_t(x) * Y_{ellipse}(x) - \frac{1}{2}\left(r_t(x)\right)^2 * (\theta - \sin\theta), \ Y_t(x) < Y_{ellipse}(x) \end{cases}$   (19)
Net areas are calculated as the differences between the areas under the sub-increment *t* and the area under the
top of the shell (sub-increment 0), and volumes for sub-increments are calculated by integrating these areas
over *x* (see formula **F7.1** above). Shell growth rates are then calculated by multiplying the change in volume per
sub-increment with *Shell Density* (*ρ*):
**F8.1**: $\Delta M_t = \rho * (V_{t-1} - V_t)$     (20)
and absolute mass accumulation is calculated by simple multiplication of the modelled incremental volume
increase of the shell with *Shell Density*:
**F8.2**: $M_t = \rho * V_t$     (21)

**2.4.2 Trace element modelling**
**Step 5** of the model takes the BMP-file of the *Phase map* of the shell and a matrix of the quantified
concentrations of all elements of interest in each of the phases as well as their colour values (*Phase data*,
**Figure 1**) as input to convert the BMP image to a matrix of phases (*Phase matrix*, **Figure 1**). This matrix is then
used to export a table with statistics of the relative abundance of phases in the entire phase map (*Phase*
*statistics*, **Figure 1**). *Phase data* tables used as input to model every shell described in this study are given in
**supplementary data 3**.



**Step 6** uses this *Phase matrix* together with *Incremental cross section* to calculate the amount of pixels of each
phase that is contained in every sub-increment (*Sub-increment phase matrix*, **Figure 1**). From this data, the
concentration of each element in each sub-increment are calculated by multiplying the relative proportion of
each phase in the sub-increment by the quantified concentrations of all elements in that phase:
$$\textbf{F9}: \boldsymbol{C_t^E} = \sum_{p=p_1}^{p=p_n} \frac{\boldsymbol{S_p}}{\boldsymbol{S_{tot}}} * \boldsymbol{C_p^E} \text{ with } \boldsymbol{p} \text{ in } [\boldsymbol{p_1}, \boldsymbol{p_2}, \boldsymbol{p_3}, \dots, \boldsymbol{p_n}] \qquad (22)$$
where $C_t^E$ is the concentration of element *E* in sub-increment *t*, *p* is the phase (in $p_1$, $p_2$, $p_3$… …$p_n$), $S_p$ is the
amount of pixels assigned to phase *p* in sub-increment *t*, $S_{tot}$ is the total amount of pixels in sub-increment *t*,
and $C_p$ represents the concentration of element E in phase p. The distribution of trace element concentrations
in each sub-increment is stored in *Matrix of concentration through time*. This matrix is then multiplied with a
smoothed record of modelled mass accumulation and growth rates (see **Step 4**, smoothing occurs via a
running average over the mass accumulation and growth rate records and the *Degree of smoothing* is
customizable and determines the window size of this running average) to calculate the cumulative
accumulation and accumulation rates, respectively, of all (trace) elements through time in the shell:
$$\textbf{F10.1}: \boldsymbol{M_t^E} = \boldsymbol{C_t^E} * \boldsymbol{M_t} \qquad (23)$$
$$\textbf{F10.2}: \left[\frac{\partial M}{\partial t}\right]_t^E = \boldsymbol{C_t^E} * \boldsymbol{N} * \boldsymbol{\Delta M_t} \qquad (24)$$
Matrices of modelled elemental concentrations (*Matrix of concentrations through time*), cumulative trace
element accumulation (*Cumulative elemental mass accumulation*) and accumulation rates (Elemental mass
accumulation rate) modelled through the shell's age are exported for further analysis. An overview of all
model functions and variables is given in **Table 1**. Exported matrices containing modelling results for every
shell featuring in this study are presented in **supplementary data 4**
**3. Results and discussion**
**3.1 XRF and shell dimension measurements**
Shells of *C. gigas* are highly irregular with considerable differences in shape between individuals, as is evident
from measurements of the shell dimensions (**Table 2**) and the colour scans of the shells (**Figure 2** and
**supplementary data**). Shell length, width, volume and mass vary considerably between *C. gigas* specimens and
estimated age based on proxy records is not a good predictor of shell size. Furthermore, the length-to-width
ratio is highly variable between shells, making size development in *C. gigas* hard to predict. Densities of *C.*
*gigas* shells are relatively low ($\rho$ = 2.10 g*cm$^{-1}$) compared to the densities of shell-forming minerals such as
calcite ($\rho$ = 2.71 g*cm$^{-1}$), aragonite ($\rho$ = 2.95 g*cm$^{-1}$) and nacre ($\rho$ = 2.60 g*cm$^{-1}$). This difference is most likely
caused by the presence of porosity in the shell structure, which should be around 23% to account for the
difference in shell density.
**Figure 2** shows the result of colour scanning, XRF mapping with phase analysis and a tracing of the growth
increments in a longitudinal cross section through one of the *C. gigas* shells. The shell depicted in **Figure 2** is



used as an example for the remainder of the results and discussion, while outcomes for the remaining seven
shells are disclosed in in **supplementary data 5**. **Figure 2** shows that phase analysis on the XRF map of oyster
shells results in the separation of four chemically distinct phases in the cross section. Comparison with the
colour scan shows that these phases represent dark foliated calcite layers in the shell (green), light chalky
calcite layers in the shell (blue), detrital inclusions in the edge of the shell (yellow) and the surrounding resin
(red). Trace element concentrations of the foliated and chalky calcite phases in each shell are found in **Table 3**.
Mapping and phase analysis in all shells resulted in a distinction between foliated calcite and chalky calcite
layers in terms of chemical composition (see **Figure 2** and compare chemical compositions in **Table 3**). The
phase map in **Figure 2** also shows that the hinge of the shell consists of foliated calcite. Traces of growth
increments in the oyster shell show once more that growth patterns in *C. gigas* are highly irregular. While shell
growth always happens by addition of material on the inside of the shell valve (facing down in **Figure 2**), shell
thickness varies strongly throughout the shell and shell extension rates vary both with age and with location in
the shell. Furthermore, oyster shells extend away from the shell hinge (to the right in **Figure 2**) as well as
towards the inside of the shell, making the hinge thicker with age (downward and to the left in **Figure 2**). These
shell characteristics complicate the modelling of shell growth and render *C. gigas* an ideal species for rigorous
testing of the model presented in this study.
Results of line scanning through the hinge of the oyster are shown in **Figure 4**. Shells of *C. gigas* are
characterized by periodic variations in concentrations of strontium (Sr), magnesium (Mg), sulphur (S), iron (Fe),
manganese (Mn) and zinc (Zn). Records of silicon (Si) and calcium (Ca) concentrations indicate which parts of
the records represent pure shell calcite (high [Ca], low [Si]) and which consist of calcite diluted with detrital
material (lower [Ca], [Si] > 2000 µg/g, mostly on the outside of the shell, see **Fig. 4**). Shell increments used as
tracers for growth modelling are generally characterized by decreased Ca and Mg concentrations and
increased concentrations of Fe, Mn, Zn and Sr. Furthermore, records of Sr and Zn show regular cyclicity, while
Fe and Mn records are characterized by sharp increases relative to a stable baseline. The Mg record shows
small scale variations inversely related to those in the Zn record. Periodic variations in the trace element
records allow the establishment of an age model for the growth of this oyster shell, as is shown in **Figure 4**.
Note that line scanning through the hinge of the shell only allows for the sampling of the last three growth
years, as the irregular shape of the oyster shell and the occurrence of chalky calcite further up the hinge
prevents the measurement of a complete record through the foliated calcite. Also note that growth
increments used as a basis for growth modelling are not paced to the seasonal cycle. The organisation of
isochronous growth increments in the colour scan on top of **Figure 4** shows the occurrence of chalky calcite
layers embedded between foliated calcite layers in some parts of the shell while these are absent in other
parts. This further confirms that multiple types of shell mineral phases (e.g. foliated calcite and chalky calcite)
can be precipitated in the shell simultaneously. Since mineral phases are chemically distinct (**Table 3**), this
observation warrants the consideration of the growth of both shell phases in an analysis of trace element
uptake by oyster shells, showing that simply analysing foliated calcite in the hinge of the shell may not yield a
complete understanding of the incorporation of trace elements into the shell.



### 3.2 Growth model

The output of the growth model applied on cross sections of *C. gigas* shells consists of records of shell length, average thickness, volume, mass and growth rates through shell age (**Figure 5**). Tables containing the complete records of all these parameters for all shells are given in **supplementary material 4**, and modelled shell dimensions at the end of the modelling run are given in **Table 2**. **Figure 5** shows the records for the above mentioned shell parameters plotted against age following the age model based on line scans through the shell hinge. The results show that, though there is ample variation in size development between individuals, the development of shell size parameters follow a similar pattern in all the examined shells. Development of shell length in all modelled shells follows the asymptotic Von Bertalanffy growth model ($L = L_\infty * e^{-kt}$; Von Bertalanffy, 1957). Parameters of Von Bertalanffy models (k and $L_\infty$) fitted to shell length records of each shell are given in **Table 2**. Results show that, while Bertalanffy curves fit the shell length development very well ($R^2 >$ 0.90 for most shells except for #3 and #4), Bertalanffy's K values (k) as well as maximum shell lengths ($L_\infty$) vary strongly. This result is unsurprising for oyster shells, which are known to show large variations in growth rate and morphology in response to local environmental constraints on their growth (Galtsoff, 1964; Palmer and Carriker, 1979). Curve fitting through a composite of all *C. gigas* shells yields a maximum shell length of 102.34 mm and the growth curve constant (Bertalanffy's K) of 0.99. The values for maximum shell length are significantly lower than the value found for sister-taxa *C. virginica* (150 mm; Rothschild et al., 1994), but this may be a result of the use of relatively young individuals in this study. The fact that the obtained results seem to fit the Von Bertalanffy model well ($R^2$ = 0.60 for all shells combined, see **Table 3** and **Figure 5**) shows that the shell growth results produced by the model are reasonable, because it is known that the Von Bertalanffy growth model describes shell length in most bivalves. The values for Bertalanffy's K fitting the model results are quite high compared to most bivalve growth studies (e.g. Bachelet, 1980; MacDonald and Thompson, 1985; Hart and Chute, 2009), but values greater than 1 are not unheard of in bivalve species that show steep growth curves early in life (e.g. Urban, 2000; Richardson et al., 2004). Modelled shell lengths closely resemble those measured on the shell, with an average offset of 0.16 mm (0.16% relative to average shell length, see **Table 2**) and are in good agreement with shell length measurements of living specimens of *C. gigas* (Diederich, 2006).

The development of other growth parameters shows similar variation within the same pattern of development between individuals of *C. gigas*, attesting to the reproducibility of the growth model. For example, the average shell thickness of oyster shells is best described by a linear increase in thickness with age (**Figure 5**). Individual results show that the initial increase in thickness (slope of the average shell thickness curve) is quite variable, but that later in life the different individuals of *C. gigas* converge towards a similar average shell thickness. This results in rather variable rates of shell thickness increase between individuals (0.54–1.61 mm/yr, see **Table 2**). The convergence of the shell thickness curves at later age suggests that this range is biased by the use in this study of relatively young individuals. These differences in the development of shell thickness in oyster shells are likely to be a result of spatial constraints on shell growth (Bartol et al., 1999). The agreement between the final thicknesses of individuals is quite remarkable given their irregular shell shape and vastly different



proportions of shell length and width (**Table 2**). Maximum thickness (thickness of the thickest part of the shell)
is not modelled and therefore cannot be compared with measured values in **Table 2**, but modelled average
thicknesses are in agreement with observations in the cross section, and are proportional to measured
maximum thickness of the shells.
Modelled shell mass and volume development are best approximated by a polynomial increase with shell age
that is in agreement with the linear increase observed in modelled growth rates of *C. gigas*, which is naturally
the derivative of shell mass development (**Figure 5**). Modelled shell volume and mass at the end of the shell's
lifetime generally underestimate measured volume and mass by 4.2 cm$^3$ and 9.0 g respectively (±21%, see
**Table 2**). The most likely reason for this offset is that the assumption of a semi-circular shape of the YZ-plane
cross-section through the shell (perpendicular to the growth axis, see **Figure 3C**) underestimates the area of
this cross section. In reality, the decrease in shell thickness towards the outer margins of the shell is probably
less pronounced. Trends in volume and growth rates are less reproducible between individuals than those in
shell length and shell thickness, as is evident from the diverging polynomial fits that fit the model data. This
behaviour illustrates the erratic growth of *C. gigas* shells, which is also evident from the shape of their shell
(**Figure 2** and **supplementary data 5**). As is shown by the modelled growth rate curves (**Figure 5**), the growth
of these oyster shells is characterized by short-lived increases in growth rate followed by periods of slower,
more constant shell growth. The implications of these periodic growth spurts punctuating more constant
growth rates are also visible in the shell volume curves that often show stepwise increases in shell volume. To
a lesser extent, the same periodic growth is seen in the records of shell length and thickness. On a closer
examination, periods of faster growth rates can be associated with either contemporary increases in shell
length or in shell thickness, but rarely both at the same time. This strongly suggests a control of available
growth space on the shape and size development of *C. gigas* shells in competition with other individuals in an
oyster reef (e.g. Palmer and Carriker, 1979; Bartol et al., 1999). On the other hand, food availability is known to
significantly affect growth rates in bivalve shells (Kerswill, 1949; Coté et al., 1994; Surge and Lohmann, 2008),
showing that peaks in growth rate found by the model results in this study may be attributed to short-lived
increases in food availability commonly associated with algal blooms in spring and autumn in the region of
study (Edwards et al., 2001; Wiltshire et al., 2008). This reliance of shell growth on environmental factors
illustrates the potential of these model results to aid in the reconstruction of environmental conditions.
**3.3 Trace element model**
Records of trace element accumulation rates and total shell trace element concentrations that result from
trace element modelling are plotted for one of the *C. gigas* shells in **Figure 6** together with concentrations in
the hinge of the shell measured using XRF line scanning. Records of accumulation rates of different elements
show similar trends during shell growth and correlate with changes in shell growth rates. For some elements
(e.g. Zn and S) the total shell concentrations over time resemble concentrations measured in the hinge of the
shell, while for other elements (e.g. Mg and Sr) the total shell concentrations show a very different pattern
from the measured concentrations in the foliated calcite in the shell hinge. The reason for this difference is
that some elements (e.g. Zn and S) have very similar concentrations in the foliated calcite and the chalky




calcite layers, whereas these concentration can be very different for other elements (e.g. Mg and Sr; see **Table**
**3**). Since the type of mineral phase deposited during shell growth is not controlled by growth seasonality or
age (Surge et al., 2001; Titschack et al., 2008), differences in the degree of incorporation of mineral phases
over time will result in different total shell concentrations. These differences in total shell concentrations and
concentrations in the shell hinge illustrate the value of the proposed trace element modelling approach, as
concentrations taken up in the shell are better reflected by total shell concentrations than by concentrations in
one of the mineral phases in the shell. Furthermore, in combining the results of trace element modelling with
measurements on the shell hinge of bivalves, it is possible to constrain the relative amount of each mineral
phase that is incorporated into the shell at any given time. This allows the reconstruction of changes in shell
mineralogy and help isolate of the factors that control these changes, which is an important question in the
study of oyster growth (e.g. Currey and Taylor, 2000; Surge et al., 2001; Titschack et al., 2008;Beniash et al.,

411 2010).

**4. Conclusions and outlook**
This study proposes a new method of modelling the growth, development and trace element incorporation in
bivalve shell based on the location of growth increments in a cross section of the shell. The advent of a
working model that can independently constrain growth and trace element uptake rates would greatly benefit
the field of bivalve sclerochronology by providing independent control on shell growth rates, which influence
the expression of geochemical proxies in the shell. This development is especially interesting for studies
dealing with extinct bivalve species for which there are no modern analogues. The basic assumptions of the
model render it applicable on all bivalve species with the same general shape and growth direction. Growth
modelling following this numerical approach yields curves of shell development with age that resemble growth
curves established via *in vivo* measurements and allows the discussion of differences in growth and
development within and between bivalve species. The present modelling approach allows the comparison of
growth and development of bivalve shells on a sub-annual scale without a priori knowledge about growth
rates in the species, opening up the comparison of proxy records in fossil bivalves with records of growth rate
derived by applying this model. This allows the discussion of the applicability of trace element concentrations
as direct tracers of environmental change as opposed to being controlled by physiological processes related to
shell growth.
The combination of growth modelling with 2D trace element XRF mapping allows the projection of trace
element distribution to a 3D model of shell volume to numerically model the total shell concentration and
accumulation of trace elements into bivalve shells. Comparison between modelled total shell trace element
content and concentrations measured along the growth axis in the shell hinge following a conventional
measurement protocol reveals different patterns in trace element concentrations. This shows that
conventional trace element profiles through the shell hinge, recording only a small part of the shell, are not
always representative for total shell concentrations and that modelling these concentrations may shed more
light on the incorporation of trace elements into bivalve calcite. Further research should therefore consist of
applying this modelling approach in other bivalve studies to compare modelled and measured trace element



concentrations. According to the results presented here, studies focusing on establishing trace element proxy
transfer functions could benefit from basing their regressions on total shell trace element concentrations
rather than measurements in the shell hinge in their attempts to isolate environmental controls on trace
element concentrations in bivalve calcite.

**Code availability**

The R script of the ShellTrace model used in this paper was published in the open source research data
repository Zenodo (http://doi.org/10.5281/zenodo.817258). The complete script used for the ShellTrace
model in this publication will be made available by means of an R package in the CRAN repository
(https://cran.r-project.org), and the script is given in **supplementary data 2**.

**Acknowledgments**

Thanks go to restaurant Jardin van Gogh in Brussels (http://www.jardinvangogh.be) for kindly supplying
the author with the pacific oysters of which the shells were used for this research project, and for
disclosing information about the origin of the bivalves. The author is financed by a personal PhD
fellowship from IWT Flanders (IWT700). Thanks go to the Hercules foundation Flanders for
acquisition of XRF instrumentation (grant HERC1309) and VUB Strategic Research for support of the
AMGC research group. The author declares that there are no conflicts of interest.

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

**FIGURE 1:** Schematic overview of the growth model and trace element module described in this paper. Yellow
boxes indicate the modelling steps (Step 1-6) described in chapter 3, diamond-shaped elements represent data
packages, ellipses represent model input parameters and boxes represent functions in the model. Elements
are connected by arrows if they interact (i.e. if data packages and/or model parameters serve as input or
output of model functions). Coloured data packages are the output of the model.
**FIGURE 2:** Example of a colour scan (top), a μXRF phase map (middle) and the digitized increments (bottom) of
a *Crassostrea gigas* shell. Phase maps show the distribution of Araldite® 2020 resin (red), foliated calcite
(green), chalky calcite (blue) and detrital material (yellow) in the shell cross sections. The *C. gigas* shell
depicted in this figure corresponds to *C. gigas* shell #1 in **Table 2**.
**FIGURE 3:** Schematic illustration of morphology of a typical bivalve shell including an indication of all
parameters used in the growth and trace element models. **Figure 2A** shows a cross section along the shell's
major growth axis (XY plane), which is the plane along which the shells were sectioned. This cross section
illustrates the parameters used to define shell increments and how interpolation between them is done (see
section 3.1, model step 2). **Figure 2B** shows an overview of the shell and a definition of the axes (X, Y and Z) as



well as the parameters defined in the base ellipse of the shell (see section 3.1, model step 3). **Figure 2C** shows
a perpendicular cross section through the width of the shell (YZ plane), which illustrates the parameters used
in the determination of shell volume (see section 3.1, model step 4).
**FIGURE 4:** Overview of the results of quantitative XRF line scanning on a *C. gigas* shell (#1 in **Table 2**). On top of
the figure is a colour scan of a cross section through the shell. The enlarged image on the left hand side shows
the shell hinge including digitized growth increments (black lines with increment numbers), annual chronology
interpreted from trace element records (yellow and transparent bands with years) and the location of the line
scan (A to B, dark blue arrow). The lower right graph shows results of trace element records along the XRF line
scan with growth increments (black lines) and annual chronology (yellow and transparent areas) indicated
vertically on the graphs. From top to bottom, records of Ca (dark blue), Si (dark red), Zn (magenta), Mn
(purple), Fe (orange), S (red), Mg (green) and Sr (blue) are plotted against line scan distance, increment
number and time on three separate x-axes at the bottom of the graph.
**FIGURE 5:** Graphs showing modelled evolution of shell length (top left), average shell thickness (bottom left),
shell volume and mass (top right) and shell growth rates (bottom right) with shell age. Solid blue lines in
different shadings indicate records from individual *C. gigas* shells. Thin dashed blue curves indicate models
fitted through the growth curves of *C. gigas* shells, while bold dashed black curves show models fitted through
a composite of modelled data from all shells combined. Regression formulae and statistics are given in **Table 2**.
**FIGURE 6:** Plotted results of trace element modelling and line scanning in one of the *C. gigas* shells (#1 in **Table**
**2**). Shaded areas indicate the evolution of modelled accumulation rates (in mg/yr) of major and trace elements
with shell age. Solid coloured lines indicate the change in modelled total shell concentrations with shell age.
Coloured points connected by black lines indicate measured elemental concentrations in the hinge of the
shells plotted against shell age (see also **Figure 4**)
**TABLE 1:** Table listing all functions used in the growth and trace element models (see chapter 3) and the
variables used in these functions. Function names and names of data packages are also found in **Figure 1** and
in the text.
**TABLE 2:** Overview of measured shell dimensions (top left), dimensions of XRF maps of all shells used in this
study (top right), shell dimensions at the end of the model run (bottom left) and parameters of growth curves
fitted through the modelled data (bottom right). Average density of shells was calculated from the averages of
shell mass and volume.
**TABLE 3:** Table listing concentrations of all elements used in this study in both chalky and foliated calcite
phases of *C. gigas* and *O. figari* shells. The "% of map"-column shows the amount of pixels the mineral phases
take up relative to the total cross section area (not including resin mapped in the XRF mapping, see **Figure 3**).



**Supplementary data 1**: SVG and CSV files of cross sections through all the *Crassostrea gigas* shells used in this
study
**Supplementary data 2**: Complete R-script used to model growth and trace element uptake as described in this
study
**Supplementary data 3**: Data of phase analysis of all trace element XRF maps including RGB colour values and
trace element concentrations of all phases.
**Supplementary data 4**: Repository containing all data matrices generated by the model ran on all shells
featuring in this study.
**Supplementary data 5**: BMP images of phase maps of all shell cross sections used as input of the trace
element model in this study.



# Figure 1



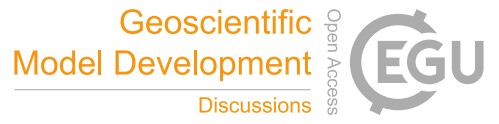

# Figure 2

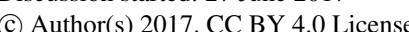

**10 mm**





**Figure 3**





# Figure 4



# Figure 5





# Figure 6



# Table 1

| Model | Model step | Function name | Equation | variables IN | source of variables | variables OUT | variables stored in: |
|---|---|---|---|---|---|---|---|
| Growth model | Step 2 | F1 | $O_i = \int_{x_0}^{x_{end}} Y_{i-1}(x) - Y_i(x)\,dx$ | $x_0$ | Digitized cross section | $A_i$ | Matrix of parameters by increment |
| | | | | $x_{end}$ | Digitized cross section | | |
| | | | | $Y_i(x)$ | Digitized cross section | | |
| | | | | $Y_{i-1}(x)$ | Digitized cross section | | |
| | | | | $dx$ | dx | | |
| | | F2 | $T_i = \frac{\sum_{x_0}^{x_{end}} Y_0(x) - Y_i(x)}{x_{end} - x_0}$ | $x_0$ | Digitized cross section | $T_i$ | Matrix of parameters by increment |
| | | | | $x_{end}$ | Digitized cross section | | |
| | | | | $Y_i(x)$ | Digitized cross section | | |
| | | | | $Y_{i-1}(x)$ | Digitized cross section | | |
| | | F3 | $L_i = \sqrt{(x_e - x_s)^2 + (Y_e - Y_s)^2}$ | $x_e$ | Digitized cross section | $L_i$ | Matrix of parameters by increment |
| | | | | $x_s$ | Digitized cross section | | |
| | | | | $Y_e$ | Digitized cross section | | |
| | | | | $Y_s$ | Digitized cross section | | |
| | Step 3 | F4.1 | $\left[\left[Y_t(x) = Y_{i-1}(x) - \frac{n}{N} * (Y_{i-1}(x) - Y_i(x))\right]_{n=0}^{n=N-1}\right]_{x=x_0}^{x=x_{end}}$ | $Y_i(x)$ | Digitized cross section | $Y_t(x)$ | Sub-incremental cross section |
| | | | | $Y_{i-1}(x)$ | Digitized cross section | | |
| | | F4.2 | $t = i - 1 + \frac{n}{N}$ | $N$ | Number of sub-increments | | |
| | | F5.1 | $a_t = \frac{1}{2} * \frac{W_{max}}{L_{max}} * (x_e - x_s)$ | $W_{max}$ | Shell width | $a_t$ | Matrix of parameters by sub-increment |
| | | | | $L_{max}$ | Shell length | | |
| | | | | $x_e$ | Sub-incremental cross section | | |
| | | | | $x_s$ | Sub-incremental cross section | | |
| | | F5.2 | $b_t = \frac{1}{2} * (x_e - x_s)$ | $x_e$ | Sub-incremental cross section | $b_t$ | Matrix of parameters by sub-increment |
| | | | | $x_s$ | Sub-incremental cross section | | |
| | Step 4 | F6.1 | $\left(\frac{Z_t(x)}{a_t}\right)^2 + \left(\frac{x^*_t}{b_t}\right)^2 = 1$ | $a_t$ | Matrix of parameters by sub-increment | $Z_t(x)$ | Z-values |
| | | F6.2 | $Z_t(x) = \left(\frac{a_t}{b_t}\right) * \sqrt{b_t^2 - x^*_t{}^2}$ | $b_t$ | Matrix of parameters by sub-increment | | |
| | | F6.3 | $x^*_t = x - x_s - b_t$ | $x$ | Sub-incremental cross section | | |
| | | | | $x_s$ | Sub-incremental cross section | | |
| | | F7.1 | $V_t = \int_{x_0}^{x_{end}} (A_0(x) - A_t(x))dx$ | $x_0$ | Sub-incremental cross section | $V_t$ | Matrix of parameters by sub-increment |
| | | | | $x_{end}$ | Sub-incremental cross section | | |
| | | | | $dx$ | dx | | |
| | | F7.2 | $Y_{ellipse} = Y_s + \left(\frac{Y_e - Y_s}{x_e - x_s}\right) * (x - x_s)$ | $Y_s$ | Sub-incremental cross section | | |
| | | | | $Y_e$ | Sub-incremental cross section | | |
| | | F7.3 | $\Delta y^2 + \Delta z^2 = r^2$ | $x_s$ | Sub-incremental cross section | | |
| | | F7.4 | $(y_1 - y_c)^2 + (z_1 - z_c)^2 = (y_3 - y_c)^2 + (z_3 - z_c)^2$ | $x_e$ | Sub-incremental cross section | | |
| | | F7.5 | $(Y_t(x) - Y_c(x))^2 + (0 - 0)^2 = (Y_{ellipse}(x) - Y_c(x))^2 + (Z_t(x) - 0)^2$ | | | | |
| | | F7.6 | $Y_c(x) = \frac{(Y_t(x)^2 - Y_{ellipse}(x)^2 - Z_t(x)^2)}{2 * (Y_t(x) - Y_{ellipse}(x))}$ | $x$ | Matrix of parameters by sub-increment | | |
| | | F7.7 | $r_t(x) = \sqrt{Z_t(x)^2 + (Y_{ellipse}(x) - Y_c(x))^2}$ | $Y_t(x)$ | Matrix of parameters by sub-increment | | |
| | | F7.8 | $\theta = 2 * \sin^{-1}\left(\frac{Z_t(x)}{r_t(x)}\right)$ | $Z_t(x)$ | Z-values | | |
| | | F7.9 | $A_t = \begin{cases} A_{segment} + A_{rectangle} = \frac{1}{2}(r_t(x))^2 * (\theta - \sin\theta) + 2 * Z_t(x) * Y_{ellipse}(x), & Y_t(x) \geq Y_{ellipse}(x) \\ A_{rectangle} - A_{segment} = 2 * Z_t(x) * Y_{ellipse}(x) - \frac{1}{2}(r_t(x))^2 * (\theta - \sin\theta), & Y_t(x) < Y_{ellipse}(x) \end{cases}$ | | | | |
| | | F8.1 | $\Delta M_t = \rho * (V_{t-1} - V_t)$ | $V_t$ | Matrix of parameters by sub-increment | $\Delta M_t$ | Matrix of parameters by sub-increment |
| | | | | $V_{t-1}$ | Matrix of parameters by sub-increment | | |
| | | F8.2 | $M_t = \rho * V_t$ | $\rho$ | Shell density | $M_t$ | Matrix of parameters by sub-increment |
| Trace element model | Step 6 | F9 | $C_t^E = \sum_{p=p_1}^{p=p_n} \frac{S_p}{S_{tot}} * C_p^E \qquad p \text{ in } [p_1, p_2, p_3, \ldots, p_n]$ | $S_p$ | Sub-increment phase matrix | $C_t^E$ | Matrix of concentration through time |
| | | | | $S_{tot}$ | Sub-increment phase matrix | | |
| | | | | $C_p^E$ | Phase data | | |
| | | F10.1 | $M_t^E = C_t^E * M_t$ | $C_t^E$ | Matrix of concentration through time | $M_t^E$ | Cumulative elemental mass accumulation matrix |
| | | | | $M_t$ | Matrix of parameters by sub-increment | | |
| | | F10.2 | $\left[\frac{\partial M}{\partial t}\right]_t^E = C_t^E * N * \Delta M_t$ | $\Delta M_t$ | Matrix of parameters by sub-increment | $\left[\frac{\partial M}{\partial t}\right]_t^E$ | Elemental mass accumulation rate matrix |
| | | | | $N$ | Number of sub-increments | | |



Geoscientific Model Development Discussions — Open Access EGU

# Table 2

**Oyster dimensions**

| # | Shell length (mm) | Shell width (mm) | Maximum shell thickness (mm) | Shell mass (g) | Shell volume (cm3) | Density |
|---|---|---|---|---|---|---|
| Crassostrea gigas #1 | 93.06 | 56.78 | 9.26 | 45.00 | 21.38 | 2.10 |
| Crassostrea gigas #2 | 100.82 | 59.18 | 7.22 | 50.46 | 23.98 | 2.10 |
| Crassostrea gigas #3 | 101.94 | 43.07 | 8.84 | 40.25 | 19.13 | 2.10 |
| Crassostrea gigas #4 | 101.46 | 53.37 | 10.62 | 39.48 | 18.76 | 2.10 |
| Crassostrea gigas #5 | 86.47 | 60.46 | 7.71 | 54.93 | 26.10 | 2.10 |
| Crassostrea gigas #6 | 83.52 | 53.95 | 3.39 | 23.88 | 11.35 | 2.10 |
| Crassostrea gigas #7 | 100.86 | 51.60 | 7.05 | 42.95 | 20.41 | 2.10 |
| Crassostrea gigas #8 | 101.50 | 57.77 | 4.54 | 39.38 | 18.71 | 2.10 |
| Average | 96.20 | 54.52 | 7.33 | 42.04 | 19.98 | |
| standard deviation | 7.53 | 5.53 | 2.40 | 9.21 | 4.38 | |

**Model results**

| # | Shell length (mm) | Shell width (mm) | Average shell thickness (mm) | Shell mass (g) | Shell volume (cm3) | Density |
|---|---|---|---|---|---|---|
| Crassostrea gigas #1 | 97.31 | 59.37 | 4.61 | 30.66 | 14.60 | 2.10 |
| Crassostrea gigas #2 | 100.45 | 58.96 | 4.89 | 38.65 | 18.40 | 2.10 |
| Crassostrea gigas #3 | 102.34 | 43.24 | 5.88 | 58.26 | 27.74 | 2.10 |
| Crassostrea gigas #4 | 101.08 | 53.17 | 6.41 | 36.57 | 17.41 | 2.10 |
| Crassostrea gigas #5 | 89.56 | 62.62 | 4.42 | 25.66 | 12.21 | 2.10 |
| Crassostrea gigas #6 | 78.80 | 50.90 | 3.83 | 15.85 | 7.55 | 2.10 |
| Crassostrea gigas #7 | 100.74 | 51.54 | 4.84 | 25.78 | 12.28 | 2.10 |
| Crassostrea gigas #8 | 100.59 | 57.25 | 5.16 | 32.97 | 15.70 | 2.10 |
| Average | 96.36 | 54.63 | 5.01 | 33.05 | 15.74 | |
| standard deviation | 8.17 | 6.18 | 0.82 | 12.47 | 5.94 | |

**Oyster XRF map dimensions**

| | map width (mm) | map length | #pixels in X-direction | #pixels in Y-direction | Total # of pixels | Spatial resolution (um) |
|---|---|---|---|---|---|---|
| Crassostrea gigas #1 | 78.10 | 24.93 | 3124 | 997 | 3114628 | 25 |
| Crassostrea gigas #2 | 101.01 | 33.30 | 3367 | 1110 | 3737370 | 30 |
| Crassostrea gigas #3 | 102.00 | 39.00 | 3400 | 1300 | 442000 | 30 |
| Crassostrea gigas #4 | 106.55 | 26.55 | 4262 | 1062 | 4526244 | 25 |
| Crassostrea gigas #5 | 88.30 | 29.50 | 3532 | 1180 | 4167760 | 25 |
| Crassostrea gigas #6 | 85.68 | 31.53 | 3427 | 1261 | 4321447 | 25 |
| Crassostrea gigas #7 | 103.00 | 20.10 | 4120 | 804 | 3312480 | 25 |
| Crassostrea gigas #8 | 102.66 | 34.80 | 3422 | 1160 | 3969520 | 30 |
| Average | | | | | 3946181 | 26.88 |
| standard deviation | | | | | 519354 | 2.59 |

**Growth curve fits**

| Fit equation | | Shell Length (mm) $L = L_0 * e^{(k*t)}$ | | | ll thickness (n) $T = a*t$ | | Shell mass (g) $M = a*t^e$ | | | Growth rate (g/yr) $dM/dt = a*t$ | |
|---|---|---|---|---|---|---|---|---|---|---|---|
| Fit parameter | N | L0 | k | R2 | a | R2 | a | e | R2 | a | R2 |
| Crassostrea gigas #1 | 700 | 140.40 | 0.28 | 0.95 | 0.80 | 0.99 | 0.20 | 3.58 | 0.98 | 2.53 | 0.62 |
| Crassostrea gigas #2 | 900 | 106.19 | 0.79 | 0.91 | 0.99 | 0.99 | 1.53 | 2.45 | 0.97 | 2.86 | 0.29 |
| Crassostrea gigas #3 | 600 | 94.76 | 1.09 | 0.25 | 0.54 | 0.93 | 0.87 | 1.97 | 0.88 | 3.45 | 0.70 |
| Crassostrea gigas #4 | 700 | 90.77 | 44.65 | 0.17 | 0.78 | 0.99 | 0.78 | 2.23 | 0.99 | 2.13 | 0.19 |
| Crassostrea gigas #5 | 700 | 91.50 | 2.29 | 0.98 | 1.61 | 0.98 | 6.58 | 2.08 | 0.88 | 4.77 | 0.32 |
| Crassostrea gigas #6 | 600 | 83.39 | 1.13 | 0.98 | 1.33 | 0.99 | 2.66 | 2.00 | 0.93 | 4.06 | 0.66 |
| Crassostrea gigas #7 | 800 | 103.28 | 1.01 | 0.91 | 0.94 | 0.99 | 0.66 | 2.76 | 0.96 | 2.44 | 0.76 |
| Crassostrea gigas #8 | 482 | 100.59 | 4.47 | 0.76 | 1.23 | 0.99 | 5.99 | 1.40 | 0.97 | 3.59 | 0.44 |
| Crassostrea gigas composite | 4655 | 102.34 | 0.99 | 0.60 | 0.94 | 0.91 | 1.83 | 1.98 | 0.76 | 2.81 | 0.21 |



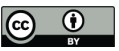

# Table 3

| | species | # | % of map | Mg (µg/g) | Si (µg/g) | P (µg/g) | S (µg/g) | Ca (µg/g) | Mn (µg/g) | Fe (µg/g) | Zn (µg/g) | Sr (µg/g) |
|---|---|---|---|---|---|---|---|---|---|---|---|---|
| Reproducibility error (1σ) | | | | ± 422 | ± 148 | ± 4 | ± 121 | ± 395 | ± 4 | ± 6 | ± 11 | ± 3 |
| **Chalky calcite** | C.gigas | 1 | 62.53% | - | 2384 | 2 | 116 | 383304 | 27 | 162 | 17 | 297 |
| | C.gigas | 2 | 68.87% | 4001 | 4259 | 137 | 5341 | 374559 | 98 | 49 | 57 | 1877 |
| | C.gigas | 3 | 59.25% | 4393 | 4934 | 151 | 5267 | 353857 | 33 | 7 | 31 | 1135 |
| | C.gigas | 4 | 49.55% | 2368 | 4239 | 44 | 2891 | 378539 | 47 | 14 | 23 | 644 |
| | C.gigas | 5 | 83.91% | 1643 | 3108 | 73 | 1627 | 390939 | 51 | 30 | 57 | 703 |
| | C.gigas | 6 | 79.12% | 1802 | 2903 | 81 | 1923 | 388243 | 66 | 43 | 27 | 611 |
| | C.gigas | 7 | 40.40% | 1847 | 4231 | 70 | 3097 | 384371 | 23 | 125 | 20 | 699 |
| | C.gigas | 8 | 71.97% | 4267 | 5175 | 182 | 5921 | 357400 | 92 | 71 | 52 | 1358 |
| **Foliated calcite** | C.gigas | 1 | 37.47% | 2744 | 4732 | 114 | 6399 | 351759 | 103 | 261 | 54 | 1433 |
| | C.gigas | 2 | 31.13% | 1153 | 3644 | 78 | 3920 | 388370 | 70 | 13 | 75 | 1432 |
| | C.gigas | 3 | 40.75% | 1801 | 3279 | 71 | 3069 | 381022 | 30 | 0 | 25 | 1382 |
| | C.gigas | 4 | 50.45% | 1304 | 1455 | 484 | 2173 | 387183 | 61 | 39 | 26 | 694 |
| | C.gigas | 5 | 16.09% | 2112 | 3837 | 48 | 1907 | 389176 | 65 | 138 | 62 | 1690 |
| | C.gigas | 6 | 20.88% | 1903 | 6047 | 81 | 2040 | 386690 | 58 | 31 | 22 | 1510 |
| | C.gigas | 7 | 59.60% | 1062 | 2161 | 31 | 1691 | 392954 | 38 | 1 | 17 | 728 |
| | C.gigas | 8 | 28.03% | 2488 | 2522 | 72 | 4077 | 380805 | 78 | 37 | 52 | 1525 |