# Peer review of "Manuscript under review for journal Geosci. Model Dev."

_Geoscientific Model Development, 2017_

## Referee Comment (RC1) · Anonymous Referee #1 · 5 Jul 2017

De Winter presented a model for bivalves that can be used to compute the amount of shell material produced in specified time intervals including its average chemistry, weight, and relative proportion of the different microstructural types (foliated, chalky, prismatic). The model was tested with oysters. The author apparently used one valve per specimen, but did not say which one, the right or left.

The manuscript is very difficult to read and I had a hard time to understand for which purpose the model is really useful. L18-20: "This approach yields records of integrated total-shell trace element concentrations and accumulation rates, which shed light on

the rates and mechanisms by which these trace elements are incorporated into the shells of bivalves." I am honestly not sure how this can be accomplished by knowing the 3D bulk chemistry deposited in selected time slices. References are needed that demonstrate that such information is relevant. L413-417: "This study proposes a new method of modelling the growth, development [= ontogenetic development of shell shape?] and trace element incorporation in bivalve shell based on the location of growth increments in a cross section of the shell. The advent of a working model that can independently constrain growth and trace element uptake rates would greatly benefit the field of bivalve sclerochronology by providing independent control on shell growth rates, which influence the expression of geochemical proxies in the shell." No reference is provided for the relationship between growth rate and shell geochemistry. Shell growth patterns are a much better tool to determine growth rates than this model. Details are given further below (critique 1).

Sentence constructions are often very complicated and the phrasing is often not concise to the point. For example (L9-10): "However, many studies have shown that physiological changes related to growth of the bivalve may overprint these chemical tracers." Why not simply: ''Vital effects overprint chemical proxies of environmental change'. Or: L295-296 "Mapping and phase analysis in all shells resulted in a distinction between foliated calcite and chalky calcite layers in terms of chemical composition" Why not simply: 'Shell portions with foliated and chalky calcite differ chemically.'

More importantly, the author does not employ the standard terminology in sclerochronology, mineralogy and malacology. A few examples: The axis of maximum growth – along which the shells were cut – agrees approximately with the height axis of the shells, but not the length axis. The latter is defined as the distance from the posterior to the anterior margin. Shell width describes the broadest distance between the two valves. The author refers to growth lines (or bands) as increments (= the distance between consecutive growth lines). Foliated, prismatic and chalky refer to microstructures, not "mineral phases" (L321). The mineral phase of bivalve shells is CaCO3 and

occurs in various different polymorphs, depending on species and shell layer or shell portion, i.e., predominantly aragonite and calcite (not just calcite as stated in line 39), as well as vaterite and amorphous calcium carbonate etc.

Aside from that, the three biggest problems of the paper are (1) potential fields of application have not been properly described, (2) initial assumptions for the model are invalid and (3) the most relevant aspect of the model, chemistry and growth rate, have not been tested by independent means.

As for critique 1 – potential fields of application. There may be interest to know how carbonate production rates and biomass changed through time, and how much $CO_2$ was sequestered by bivalves and other organisms. This could be accomplished by the analysis of shell volumes produced in specified time intervals. However, de Winter placed paleoclimate research and bivalve sclerchronology in the foreground without actually detailing how precisely these fields can benefit from his model. For example, the chemistry in a specified volume of a microstructurally and thus, chemically highly heterogeneous shell is – in my view – of limited relevance for paleoclimate research. The formation of different microstructures may be environmentally controlled, but are also governed by biomechanical needs. Since the different microstructures vary chemically (also mentioned in the ms, but no citations provided), chemical analyses should be limited to the same shell layer and microstructure. The author needs to explain and support by references why he thinks that the chemistry of a contemporaneously formed shell volume is relevant.

To assign dates to a shell, to determine how much shell material was deposited each day (L14) or month, the shell growth patterns can be consulted (note, no "proxy" is needed to "estimate age": L281). In fact, daily growth lines have been reported from this species (e.g., Barbin et al. 2008 DOI 10.1007/s00531-006-0160-0) and could have been used to determine the duration of the growing season and assign (calendar) dates to any portion of the shell. It is relevant to point out that no bivalve grows uninterruptedly through the year, because otherwise no growth lines (= reflecting periods of

retarded and ceased growth) would have been developed! One will certainly find less than 365.25 daily increments in an annual growth increment of a modern shell. In addition, the width of these daily increments provides information on the seasonal rate of shell formation which most certainly differs from the seasonally resolved growth curves shown in Figure 5 (upper left). Unfortunately, de Winter based his age model only on annual growth patterns and regular oscillations in some element curves (L331), which he interpreted as seasonal ("periodic": L306) variations. I wish to point out that there is still no consensus on the use of trace and minor elements in bivalves as environmental proxies, and I was unable to find any evidence in the manuscript supporting the assertion that the element levels in the studied specimens were actually controlled by the environment. The Van Bertalanffy curve may be able to approximate the cumulative annual shell growth, but the portions between the points in Figure 5 should not be over-interpreted and used to obtain a "daily" (L14) resolved shell growth record. Interestingly, the term "daily" is only used in the Abstract, but nowhere else in the manuscript. Instead, the author used the term "sub-annual" (L423) and also introduced the term "sub-increments" without explaining what this actually means. Is this referring to the differences in brightness resulting from the foliated and chalky microstructures?

As for 2 – false initial assumptions. I agree that a model approximates the real world, but may not perfectly resemble it. One may accept for a moment that the model reconstructed mass and volume with an error of 21%, which is attributed to variable shell thickness, specifically toward the posterior and anterior margins, possible ontogenetic changes in microstructure, organic content and porosity etc. But what also varies toward the anterior and posterior sides is the shell chemistry (e.g., Carriker et al 1991 Marine Biology 109, 287-297), and I dare to predict that the microstructure (including the relative proportion of foliated, prismatic and chalky microstructures etc.) varies in these directions as well, and with it does the chemistry (and the density). Interpolating the chemistry in a volume of contemporaneously precipitated shell portion from a single (2D) cross-section is therefore not valid; the chemical and structural heterogeneity is too large and cannot be modeled as presented.

As for 3 – model not tested. The paper would gain more strength if the model outcome was tested (not just mass and weight). In particular, daily growth patterns in the cross-section could have been used to test how well the Van Berlalanffy curve estimated varying seasonal growth rates. To test how well the volume of shell (deposited in selected time intervals) was estimated, the shells should be cut in various different angles. Computer tomography would provide even more reliable volumetric data and may also give information on spatial changes of the shell microstructure. In addition, the chemistry must be tested in different portions of the shells, particularly toward the posterior and anterior ends. And if the application in paleoclimate research stays in the foreground, then the 3D chemical data must be compared to instrumental environmental records. This certainly also requires a robust temporal alignment of the shell record which daily growth pattern analysis can provide.

In conclusion, the paper needs rigorous rewriting and refocusing. Standard terminology must be adhered to. A native speaker must read the ms.

Selected other issues, more provided in annotated pdf: In various places I missed proper references. For example, L65+66: "All these physiological changes, such as variations in growth and metabolic rate, shell mineralogy and spawning events, which affect the incorporation of trace elements into the shell of bivalves, complicate the use of trace element records to complement environmental reconstruction by stable isotope sclerochronology, (Klein et al., 1996b; Gillikin et al., 2005; Immenhauser et al., 2005; Freitas et al., 2006)." None of the cited articles refer to spawning-controlled element load.

L77: Paper "under review" cannot be cited.

L116-118: "Errors of reproducibility of $\mu$XRF measurements are generally higher than the instrumental error and depend on the integration time and the excitation energy of the element (see de Winter and Claeys, 2017; de Winter et al., 2017)." Is there really no reference predating 2017 that arrived at the same conclusion?

L122-123: "Timing of shell deposition was inferred from annual cyclicity in trace element profiles." [cyclicality!] No reference provided for this assertion.

L95/96: Shells were "sectioned longitudinally along their axis of maximum growth using a slow rotating, diamond coated saw".

Reference list is done sloppy. Genus and species must be italicized. In "$\delta$18O" 18 must be superscript. Nouns in journal names must be capitalized.

Figure 2: Meaning of colors in XRF scan?

Figure 4: Size of letters varies too much (scales and axes' numbers). Why are annual growth lines not contouring the microstructure (compare e.g., Mouchi et al doi:10.1007/s00227-016-3040-6). Why are some years in enlarged hinge image flipped by 180°? Use "$\mu$" not "u". Scale bars are far too thick. X-axis of graph must be mirrored (most recent year must be to the right, time axis should have same orientation as shell in topmost figure. Error bars missing.

Figure 6: Purpose of plotting 2D shell again? You are showing 3D chemical data... Numbers on axes are far too small. Green is not legible. Do not use colors for axes' labels. Where are the error bars?

Please also note the supplement to this comment: https://www.geosci-model-dev-discuss.net/gmd-2017-137/gmd-2017-137-RC1-supplement.pdf

---

## Referee Comment (RC2) · Anonymous Referee #2 · 17 Jul 2017

Review of de Winter "ShellTrace v1.0 - A new approach for modelling growth and 1 trace element uptake in marine bivalve shells: Model verification on pacific oyster shells (Crassostrea gigas)"

This manuscript evaluates the trace elemental composition of the shells of the pacific oyster using an integrated proxy and modelling approach. There is currently significant debate in the peer-reviewed literature regarding the use of trace elemental ratios derived from marine bivalve molluscs as palaeoenvironmental proxies. This debate is largely associated to developing an understanding of the contribution of environmental conditions and vital effects have on the trace elemental composition. These effects have been reported to create significant heterogeneity in the trace elemental composition of bivalve calcium carbonate and thus developing a robust quantitative understanding of these effects is required if we are to reach a stage where trace elemental ratios could be applied in a robust palaeoenvironmental application. Therefore a detailed study utilising modelling and actual shell data is very interesting.

This study takes an interesting approach to developing an understanding of the relationships between shell growth metrics, age and trace elemental composition. I think that approaches undertaken in this manuscript could have the potential to significantly develop our understanding of the drivers of trace elemental composition in oyster shells and potentially other long-lived marine bivalves, if more widely applied. However, as the manuscript is currently presented there appear to be significant flaws in the methodology that lead to the stated conclusions being unsupported by the presented data. I have serious misgivings about the assumptions that appear to have been applied in the development of the model and in the manner in which the data are presented. The manuscript therefore needs significant revisions before it reaches a publishable standard.

General concerns:

My main concern revolves around the generation and interpretation of trace element data, and the lack of an independent age model based on the interpretation of growth increment patterns. Firstly, the line used for sampling the trace elements, as shown in Figure 4, does not follow the axis of maximum growth (which is the conventional approach for sampling geochemistry in shells) and is not perpendicular to any of the growth increments sampled. This means that the temporal averaging of the trace elemental ratios in each analyses in each increment is inconsistent. This makes it very difficult to compare data between increments. Other complications such as the potential of non-linear seasonal growth rates are not considered. It is not clear from the data that are presented that there are any consistent seasonal patterns in the elemental

ratios. The author must provide a statistical assessment of the mean seasonal trend for each element analysed. As this is not clear it is near impossible to evaluate if the relationships between elemental ratios and age is robust. I go through each of these points in more detail below.

Detailed comments:

Line 38-39 – This is a strange list of references to use. There are many more papers that arecurrently already published that would be better to cite here. For example (Butler et al., 2013, Butler et al., 2011, Reynolds et al., 2016, Wanamaker et al., 2008a, Wanamaker et al., 2008b, Wanamaker et al., 2011, Schone et al., 2003, Schöne et al., 2011, Jones, 1980, Jones et al., 1989, Witbaard et al., 2005, Witbaard et al., 1997, Witbaard et al., 2003, Witbaard et al., 1994, Swart et al., 2010)

Lines 41-42 – The author needs to cite a better range of literature.

Line 47 – should not cite work that is not published (remove deWinter et al., PPP)

Line 39 – not all bivalves are calcite, change to calcium carbonate. This term bivalve calcite is used throughout the manuscript and should be changed.

Lines 50-52 it would be useful for the author to expand on this statement to say exactly which trace elements have been used to examine which environmental archives and importantly from which bivalve species these studies were conducted. MY experience is that there is a lot of contradictory literature surrounding the application of trace elements as environmental proxies and so a far more detailed discussion around this is needed.

Lines 53-59 – Again the author needs to provide a far more detailed discussion of what these previous studies into oyster shells found. "There is some discussion about the role of these calcite mineral phases, whether their precipitation is controlled by environmental conditions and whether changes in the precipitated mineral phase are paced to regular (solar or lunar) cycles" So what do they actually say in this discussion?

What environmental factors do they suggest are responsible? Were there any issues identified in these studies? The author needs to expand significantly here.

69-71 – The author needs to be more concise in the structuring of sentences throughout the manuscript. As it is currently written it is quite difficult to read or. One example for sentences that could be reduced is: "Furthermore, several studies have shown that rates by which trace elements are incorporated into bivalve shells and the degree to which these rates are controlled by environmental factors can be vastly different between different bivalve species". This could be condensed to, for example, "Studies have shown that the rates of incorporation, and environmental response of, trace elements in bivalve shells varies between species"

77-85 – this section reads far too much like a methods summary and not a set of aims or hypotheses that this study aims to test. This should be rephrased.

88 – I presume not, but please state whether the shells cooked by the restaurant before they were analysed in this study? If so would this have had an impact on the elemental composition of the shell?

90 - from what water depth were the shells collected?

105 – clarify that you mean the polished shell surface.

112 – "using the point-by-point scanning method outlined in de Winter et al. (in review, PPP)" Firstly you shouldn't cite work that is not published. Secondly we cannot see what this method is as it hasn't been published. We therefore cannot assess if this is a robust methodology. The author should elaborate on the details of this methodology.

281 – why would you estimate age based on proxy records? Growth increment analyses are a far more robust measure of ontogeny.

309-311 - "Shell increments used as tracers for growth modelling are generally characterized by decreased Ca and Mg concentrations and increased concentrations of Fe, Mn, Zn and Sr." This is not obvious from Fig 4 at all. I personally cannot see this interpretation in these data the way they are plotted. If there is a consistent response in the elemental ratios throughout the year, with either peaks or troughs corresponding to the growth lines then the authors should generate a mean seasonal elemental curve using these data and plot with error bars the mean seasonal change in each element. This will then provide a more rigorous assessment of whether the ratios do indeed show a change occurring around the timing of the growth check forming. However, I am concerned by the trajectory that was used to generate these data as shown in Figure 4 (line A-B). The XRF sampling line (line A-B) does not follow the axis of maximum growth through the hinge region of the shell (the maximum growth appears to follow the position of the numbers 0-7). The position of the sampling line as is not perpendicular to any of the growth increments meaning the temporal positioning of each set of analyses through these increments is not consistent. This will lead to large changes in the temporal smoothing of the data through the timeseries and lead to significant challenges in interpreting the data. For instance this means that comparing one year to the next is very difficult as they incorporate a different temporal smoothing. The plotting of these data against a constant time axis is therefore inappropriate. Plotting the data like this makes the assumption that the growth of these shells is constant throughout the growing season and that the growing season is 12 months. Both of these assumptions are likely false as has been found in many other species of bivalve. The author needs to provide a far more robust approach to determining the timing and rate of seasonal growth before these data can be plotted against a time axis.

Fig 4: This figure is extremely complicated to understand. The impression this figure gives me is that the sampling strategy of this study is fundamentally flawed. For example, the black lines, which apparently correspond to the growth increments show there are 7 increments yet these do not appear to line up with annual chronology in any way? Why is this? Also the panels on this figure need to be labelled and referenced more clearly in the fig legend.

312-313 – this needs to be tested.

313-314 – why did the authors use the elemental ratios to determine an age model when the growth increments would have provided a far more robust, and independent age model? Based on the data that is displayed in Figure 4 it is extremely difficult to see any consistent patterns that would facilitate the attribution of an age model with any degree of confidence. As mentioned above, if the elemental ratios are to be used as an age model, the data from each increment should be averaged to generate a mean seasonal curve for each element. These data, with associated error bars could potentially be used to assign ages, providing the data demonstrate a significant seasonal pattern.

315+317 – "Note that line scanning through the hinge of the shell only allows for the sampling of the last three growth years, as the irregular shape of the oyster shell and the occurrence of chalky calcite further up the hinge prevents the measurement of a complete record through the foliated calcite" The author needs to explain why this line was used for sampling and why the line that follows the position of the numbers in Figure 4, lower left panel, which contains 7 growth increments wasn't used. Using this line would have facilitated sampling through 7 increments.

326 – Growth model section – I really don't understand why the author has not used the visible growth increments to develop the age model and instead used the trace elemental records. I am not convinced that the geochemical signatures provide a robust signal for age model development based on the data that has been presented in this manuscript. The author needs to provide significant new data comparing growth increment based age models in relation to the geochemical records before these analyses can be assessed in a robust manor.

344 – How does the Von Vertalanffy R2 statistic (0.60) compare to that of other bivalves?

391- Trace element model section. In this section the author presents "Records of trace element accumulation rates and total shell trace element concentrations" and

discusses these data based on Figure 6 which plots these data against a time axis over the period from 2014-2017. As mentioned with Figure 4, these analyses assume a constant seasonal growth rate and that growth occurs all year round. Yet no evidence or citations are provided to support this claim.

Figure 6 – remove the scan of the shell in the lower panel as it is not needed. Also all the panels should be labelled (A, B etc.)

403-404 - "These differences in total shell concentrations and concentrations in the shell hinge illustrate the value of the proposed trace element modelling approach" To be honest I am not sure why this is important. In the application of sclerochronological trace elemental ratios we are looking to explain the patterns of variability that we can measure along a 1 or 2 dimensional growth axis in the shell. Whilst it is of interest to understand the reasons behind heterogeneity in trace elemental ratios across the sampling plane, we are rarely interested in total shell elemental composition. The big question that needs addressing with trace elemental work is to what degree is the variability in trace elemental ratios, sampled along the axis of maximum growth, due to vital effects and what proportion is due to changes in ambient water chemistry. The author does hint at this in lines 406-411 however, I feel that this has not been achieved by this study.

418-419 – I strongly disagree with this statement. There is no evidence that this is the case presented in this study and should be removed.

422-424 – This statement if wrong. The assumptions made in this study if applied to other marine bivalves such as those commonly used in sclerochronology (e.g. Arctica islandica) would likely not hold. This study assumes constant annual growth which is not the case for A. islandica.

Suggested references

Butler, P. G., A. D. Wanamaker, Jr., J. D. Scourse, C. A. Richardson and D. J. Reynolds

(2011). Long-term stability of delta c-13 with respect to biological age in the aragonite shell of mature specimens of the bivalve mollusk arctica islandica. Palaeogeography Palaeoclimatology Palaeoecology, 302, 21-30. Butler, P. G., A. D. Wanamaker, J. D. Scourse, C. A. Richardson and D. J. Reynolds (2013). Variability of marine climate on the north icelandic shelf in a 1357-year proxy archive based on growth increments in the bivalve arctica islandica. Palaeogeography, Palaeoclimatology, Palaeoecology, 373, 141-151. Jones, D. S. (1980). Annual cycle of shell growth increment formation in 2 continental-shelf bivalves and its paleoecologic significance. Paleobiology, 6, 331-340. Jones, D. S., M. A. Arthur and D. J. Allard (1989). Sclerochronological records of temperature and growth from shells of mercenaria-mercenaria from narragansett bay, rhode-island. Marine Biology, 102, 225-234. Reynolds, D. J., J. D. Scourse, P. R. Halloran, A. J. Nederbragt, A. D. Wanamaker, P. G. Butler, C. A. Richardson, J. Heinemeier, J. Eiriksson, K. L. Knudsen and I. R. Hall (2016). Annually resolved north atlantic marine climate over the last millennium. Nature Communications, 7. Schone, B. R., W. Oschmann, J. Rossler, A. D. F. Castro, S. D. Houk, I. Kroncke, W. Dreyer, R. Janssen, H. Rumohr and E. Dunca (2003). North atlantic oscillation dynamics recorded in shells of a long-lived bivalve mollusk. Geology, 31, 1037-1040. Schöne, B. R., A. D. Wanamaker, J. Fiebig, J. Thébault and K. Kreutz (2011). Annually resolved $\delta$13cshell chronologies of long-lived bivalve mollusks (arctica islandica) reveal oceanic carbon dynamics in the temperate north atlantic during recent centuries. Palaeogeography, Palaeoclimatology, Palaeoecology, 302, 31-42. Swart, P. K., L. Greer, B. E. Rosenheim, C. S. Moses, A. J. Waite, A. Winter, R. E. Dodge and K. Helmle (2010). The c-13 suess effect in scleractinian corals mirror changes in the anthropogenic co2 inventory of the surface oceans. Geophysical Research Letters, 37. Wanamaker, A. D., Jr., J. Heinemeier, J. D. Scourse, C. A. Richardson, P. G. Butler, J. Eiriksson and K. L. Knudsen (2008a). Very long-lived mollusks confirm 17th century ad tephra-based radiocarbon reservoir ages for north icelandic shelf waters. Radiocarbon, 50, 399-412. Wanamaker, A. D., K. J. Kreutz, B. R. Schoene, Pettigrew.N., H. W. Borns, D. S. Introne, D. Belknap, K. A. Maasch and S. Feindel (2008b). Coupled north atlantic slope water

forcing on gulf of maine temperatures over the past millennium. Climate Dynamics, 31, 183-194. Wanamaker, A. D., K. J. Kreutz, B. R. Schöne and D. S. Introne (2011). Gulf of maine shells reveal changes in seawater temperature seasonality during the medieval climate anomaly and the little ice age. Palaeogeography, Palaeoclimatology, Palaeoecology, 302, 43-51. Witbaard, R., G. C. A. Duineveld, T. Amaro and M. J. N. Bergman (2005). Growth trends in three bivalve species indicate climate forcing on the benthic ecosystem in the southeastern north sea. Climate Research, 30, 29-38. Witbaard, R., G. C. A. Duineveld and P. Dewilde (1997). A long-term growth record derived from arctica islandica (mollusca, bivalvia) from the fladen ground (northern north sea). Journal of the Marine Biological Association of the United Kingdom, 77, 801-816. Witbaard, R., E. Jansma and U. Sass Klaassen (2003). Copepods link quahog growth to climate. Journal of Sea Research, 50, 77-83. Witbaard, R., M. I. Jenness, K. Vanderborg and G. Ganssen (1994). Verification of annual growth increments in arctica islandica l from the north-sea by means of oxygen and carbon isotopes. Netherlands Journal of Sea Research, 33, 91-101.

---

## Author Comment (AC1) · 24 Jul 2017

Dear Editor, Referees and contributors,

I would like to express my thanks to both anonymous referees for their careful and thorough review of my manuscript under discussion in Geoscientific Model Development. After careful consideration of the comments and suggestions posed by the referees, I present below a short breakdown of the major points of critique and a summary of suggested ways in which the manuscript may be revised. Appended to this reply, I

provide two documents containing a point by point rebuttal to all comments raised by both referees.

Points of major criticism:

1. Aim of the model

Both referees have expressed doubts about the goal of the study and more specifically about the aim of the growth and trace element model that is presented. In the point-by-point rebuttal I provide more details illustrating how the modelled total shell trace element concentrations and accumulation rates can aid in the discussion about the use of trace element concentrations in bivalve shell carbonate for the reconstruction of palaeoenvironmental conditions. In a revised version of the manuscript, I plan to rewrite the Introduction to more clearly explain the goals of the study and the added value of the data that can be gathered by implementing the presented model. In the process, I hope to make the manuscript easier to read by providing a better structure.

2. Sentence construction and use of English language

I acknowledge the criticism by both referees stating that the formulation of some sentences throughout the manuscript makes the text hard to read. In a revised version, I will go through the manuscript text in detail in order to clarify these sentences and improve the legibility of the manuscript.

3. Terminology

As both referees point out, the incorrect use of mineralogical and sclerochronological terminology can be confusing the in current version of the manuscript. In a revised version, care will be taken to adhere to the proper terminology while describing both morphological and mineralogical characteristics of bivalve shells.

4. Age model

Both referees raise fair criticism about the way in which age models are created for XRF

line scan data and model results. In a revised version of the manuscript, an attempt will be made to incorporate measurements of daily growth increments of the shells to establish an independent chronological framework for model results and data in the study.

5. Sub-annual model results

Both reviewers have doubts concerning the claims that the model presented in this manuscript is able to reconstruct growth and trace element uptake rates on a sub-annual scale. It is my hope that a revision of the model that implements growth stops and seasonal variation in growth rates in the shells will improve the reliability of the model results and reflect the natural growth patterns of marine bivalve shells more accurately.

6. Independent model verification

Concerns were raised about the degree of model verification that is applied to test the reliability of the model results discussed in this study. I agree to this point of critique and will solve this issue by adding additional measurements of the oyster's dimensions and compare them to the 3D model results. Furthermore, chemical analyses of more different parts of the oyster shells will be used to test the assumptions of the model concerning the extrapolation of shell size and heterogeneity in the anterior-posterior direction of the shell. Please refer to the point-by-point rebuttal of Referee #1's comments attached to this reply.

7. XRF measurement strategy

Both referees expressed concern about the strategy by which trace element profiles that were compared to modelling results were measured in this study. In the point-by-point rebuttal of Referee #2's comments, a more detailed explanation is given for my choice of this sampling strategy. However, in a revision of the manuscript, suggestions by Referee #2 will be taken into account and a more thorough discussion of the

sampling strategy and a comparison between results from different line scans on the polished shell surface will be added. I hope that this discussion of sampling protocol will shed more light on the heterogeneity within the bivalve shell and that the comparison of the different line scan results with the model output will allow for a better discussion of the added value of the presented model.

8. Discussion of model results in terms of physiological and environmental parameters

Some concerns were raised with reference to the discussion of the modelling results and their comparison with conventional trace element line scans. In a revised version, this part of the discussion will be rewritten to include a more careful discussion of the comparison between measured and modelled data. I will attempt to structure this part of the discussion more clearly and provide more literature support to enable the discussion of the use of modelled data to isolate the effects of physiology on trace element incorporation into the shell.

I hope that the above provides a clear outline of the revisions that I propose in reply to the comments by both referees. In addition, I invite both referees, the topical editor and other interested members of the community to express their opinion about my reply to the concerns summarized above, or to post additional comments with reference to the current version of my manuscript by participating in the online discussion. I would much appreciate any further feedback that will help me improve both the bivalve growth and trace element model and the manuscript.

Kind regards,

Niels de Winter

Analytical and Environmental Geochemistry (AMGC) Research Group

Vrije Universiteit Brussel

Brussels, Belgium

[Figure]

Please also note the supplement to this comment:
https://www.geosci-model-dev-discuss.net/gmd-2017-137/gmd-2017-137-AC1-supplement.pdf

————————————————
[Figure]

**Supplement:**

**Reply to comments raised by Anonymous Referee #1**

Dear Editor and Referees,

Please find below a point-by-point reply to the comments and remarks posted by Referee #1 on my manuscript titled "ShellTrace v1.0 – A new approach for modelling growth and trace element uptake in marine bivalve shells: Model verification on pacific oyster shells (*Crassostrea gigas*)", which was submitted for interactive discussion to Geoscientific Model Development (gmd-2017-137)

*De Winter presented a model for bivalves that can be used to compute the amount of shell material produced in specified time intervals including its average chemistry, weight, and relative proportion of the different microstructural types (foliated, chalky, prismatic). The model was tested with oysters. The author apparently used one valve per specimen, but did not say which one, the right or left.*

This is a valid comment, and details will be provided about which shell valve is used in a revised version of the manuscript

*The manuscript is very difficult to read and I had a hard time to understand for which purpose the model is really useful. L18-20: "This approach yields records of integrated total-shell trace element concentrations and accumulation rates, which shed light on the rates and mechanisms by which these trace elements are incorporated into the shells of bivalves." I am honestly not sure how this can be accomplished by knowing the 3D bulk chemistry deposited in selected time slices. References are needed that demonstrate that such information is relevant. L413-417: "This study proposes a new method of modelling the growth, development [= ontogenetic development of shell shape?] and trace element incorporation in bivalve shell based on the location of growth increments in a cross section of the shell. The advent of a working model that can independently constrain growth and trace element uptake rates would greatly benefit the field of bivalve sclerochronology by providing independent control on shell growth rates, which influence the expression of geochemical proxies in the shell." No reference is provided for the relationship between growth rate and shell geochemistry. Shell growth patterns are a much better tool to determine growth rates than this model. Details are given further below (critique 1).*

This major concern of Referee #1 is acknowledged and will be addressed in a new version of the manuscript. I understand that the introduction of the model in the manuscript in the current version lacks clarity. Therefore, the aim of the model may have been understated. This issue is clear from the comment above as well as from various other comments further in the report of Referee #1, as well as in comments of Referee #2. In a revised version of the manuscript, I will try to better explain the purpose of the bivalve growth and trace element model and how the community can benefit from this modelling attempt. More details will be provided below.

*Sentence constructions are often very complicated and the phrasing is often not concise to the point. For example (L9-10): "However, many studies have shown that physiological changes related to growth of*

*the bivalve may overprint these chemical tracers." Why not simply: ''Vital effects overprint chemical proxies of environmental change'. Or: L295-296 "Mapping and phase analysis in all shells resulted in a distinction between foliated calcite and chalky calcite layers in terms of chemical composition" Why not simply: 'Shell portions with foliated and chalky calcite differ chemically.'*

I thank Referee #1 for pointing this out, and I agree that the manuscript can be improved significantly by simplifying sentences and improving the level of English used in the text. This will require special attention in further revision of the manuscript.

*More importantly, the author does not employ the standard terminology in sclerochronology, mineralogy and malacology. A few examples: The axis of maximum growth – along which the shells were cut – agrees approximately with the height axis of the shells, but not the length axis. The latter is defined as the distance from the posterior to the anterior margin. Shell width describes the broadest distance between the two valves. The author refers to growth lines (or bands) as increments (= the distance between consecutive growth lines). Foliated, prismatic and chalky refer to microstructures, not "mineral phases" (L321). The mineral phase of bivalve shells is CaCO3 and occurs in various different polymorphs, depending on species and shell layer or shell portion, i.e., predominantly aragonite and calcite (not just calcite as stated in line 39), as well as vaterite and amorphous calcium carbonate etc.*

I apologize for the confusion that has been caused by these mistakes in the use of standard terminology. In a revised version, the correct use of sclerochronological and mineralogical terms will receive special attention. I understand that this will make the manuscript, especially the sections dealing with shell morphology, much easier to read.

*Aside from that, the three biggest problems of the paper are (1) potential fields of application have not been properly described, (2) initial assumptions for the model are invalid and (3) the most relevant aspect of the model, chemistry and growth rate, have not been tested by independent means.*

*As for critique 1 – potential fields of application. There may be interest to know how carbonate production rates and biomass changed through time, and how much CO2 was sequestered by bivalves and other organisms. This could be accomplished by the analysis of shell volumes produced in specified time intervals. However, de Winter placed paleoclimate research and bivalve sclerchronology in the foreground without actually detailing how precisely these fields can benefit from his model. For example, the chemistry in a specified volume of a microstructurally and thus, chemically highly heterogeneous shell is – in my view – of limited relevance for paleoclimate research. The formation of different microstructures may be environmentally controlled, but are also governed by biomechanical needs. Since the different microstructures vary chemically (also mentioned in the ms, but no citations provided), chemical analyses should be limited to the same shell layer and microstructure. The author needs to explain and support by references why he thinks that the chemistry of a contemporaneously formed shell volume is relevant.*

This is an important point raised by Referee #1 which, in my opinion, illustrates that the aim of the model has not been properly addressed in the Introduction. I agree with Referee #1 that shell production rates are in itself interesting outcomes of the model. I also agree that microstructural heterogeneity in bivalve shells implicates chemical heterogeneity. Indeed, the variation in chemical signature of different shell microstructures is demonstrated in this manuscript. However, I also agree that this is not the novelty of the paper (as stated in comments below). The main aim of the model presented in this manuscript is to

trace the (absolute) amounts of different trace elements that are built into the bivalve shell at any given moment in time. Such information cannot be obtained by limiting the analysis of shell material to one microstructure, exactly because different microstructures are deposited at the same moment of shell growth. The rationale of trying to reconstruct total shell concentrations and accumulation rates is that these parameters include every microstructure in the shell and therefore give a more complete picture of the fluxes of trace elements into the shell. It is my hope that this additional information, together with information about the chemistry of individual shell microstructures might shed more light on the mechanisms of incorporation of these trace elements in the shell. In the future, this may improve the usefulness of trace element proxies in reconstructing palaeoenvironmental conditions from bivalve shells. In a revised version, the explanation of this goal will be improved and moved to the Introduction to provide the reader with a better understanding of the aim of the study.

*To assign dates to a shell, to determine how much shell material was deposited each day (L14) or month, the shell growth patterns can be consulted (note, no "proxy" is needed to "estimate age": L281). In fact, daily growth lines have been reported from this species (e.g., Barbin et al. 2008 DOI 10.1007/s00531-006-0160-0) and could have been used to determine the duration of the growing season and assign (calendar) dates to any portion of the shell. It is relevant to point out that no bivalve grows uninterruptedly through the year, because otherwise no growth lines (= reflecting periods of retarded and ceased growth) would have been developed! One will certainly find less than 365.25 daily increments in an annual growth increment of a modern shell. In addition, the width of these daily increments provides information on the seasonal rate of shell formation which most certainly differs from the seasonally resolved growth curves shown in Figure 5 (upper left). Unfortunately, de Winter based his age model only on annual growth patterns and regular oscillations in some element curves (L331), which he interpreted as seasonal ("periodic": L306) variations. I wish to point out that there is still no consensus on the use of trace and minor elements in bivalves as environmental proxies, and I was unable to find any evidence in the manuscript supporting the assertion that the element levels in the studied specimens were actually controlled by the environment. The Van Bertalanffy curve may be able to approximate the cumulative annual shell growth, but the portions between the points in Figure 5 should not be overinterpreted and used to obtain a "daily" (L14) resolved shell growth record. Interestingly, the term "daily" is only used in the Abstract, but nowhere else in the manuscript. Instead, the author used the term "sub-annual" (L423) and also introduced the term "sub-increments" without explaining what this actually means. Is this referring to the differences in brightness resulting from the foliated and chalky microstructures?*

The concern raised by Referee #1 is valid and shows that, indeed, the timewise interpolation done in the model may be an oversimplification. The model could indeed benefit from a consideration of seasonal changes in growth rate and growth stops to allow for more accurate interpolation of growth rates between annual growth lines. I appreciate the suggestion of applying the position of daily growth increments to independently constrain sub-annual chronology in the shell. The interpolation between growth lines implemented in the model could be improved using the result of relative thicknesses of daily growth increments or a sinusoidal model of seasonality (including growth stops). This would improve the reliability of sub-annual chronology of the model result. In a revised version of the model and the manuscript, an attempt will be made to incorporate either of these constraints on the interpolation step of the model. Note that the term "sub-increment" is used in the model description for one interpolated step between two growth lines, and that its size depends on an arbitrary "time" resolution (amount of steps between each set of growth lines) that can be changed by changing the

"Increment" parameter of the model. I agree that a name like "sub-line" might be better for this element in the model.

*As for 2 – false initial assumptions. I agree that a model approximates the real world, but may not perfectly resemble it. One may accept for a moment that the model reconstructed mass and volume with an error of 21%, which is attributed to variable shell thickness, specifically toward the posterior and anterior margins, possible ontogenetic changes in microstructure, organic content and porosity etc. But what also varies toward the anterior and posterior sides is the shell chemistry (e.g., Carriker et al 1991 Marine Biology 109, 287-297), and I dare to predict that the microstructure (including the relative proportion of foliated, prismatic and chalky microstructures etc.) varies in these directions as well, and with it does the chemistry (and the density). Interpolating the chemistry in a volume of contemporaneously precipitated shell portion from a single (2D) cross-section is therefore not valid; the chemical and structural heterogeneity is too large and cannot be modeled as presented.*

This heterogeneity in anterior-posterior direction is indeed not accounted for in the model. I acknowledge that, if a cross section through the shell in direction of maximum height is not representative of the entire shell, the volume and trace element concentrations given by the model will deviate from the real values. However, the only way to work around this problem is to measure chemical composition and density in three dimensions, which defeats the purpose of this simple, accessible and inexpensive model approach. As correctly stated by Referee #1, this model remains an approximation of real life conditions and has its limitations. However, the high resolution (25 µm x 25 µm) that can be achieved with XRF mapping ensures that the chemical heterogeneity in the 2D cross section can be mapped with high precision. The statistics of the ±4 million pixels in each map may ensure that the heterogeneity within the shell is well represented in the 2D section. However, if the anterior and posterior extremes of the shell contain consistently more of one particular microstructure, this will affect model results.

*As for 3 – model not tested. The paper would gain more strength if the model outcome was tested (not just mass and weight). In particular, daily growth patterns in the cross-section could have been used to test how well the Van Berlalanffy curve estimated varying seasonal growth rates. To test how well the volume of shell (deposited in selected time intervals) was estimated, the shells should be cut in various different angles. Computer tomography would provide even more reliable volumetric data and may also give information on spatial changes of the shell microstructure. In addition, the chemistry must be tested in different portions of the shells, particularly toward the posterior and anterior ends. And if the application in paleoclimate research stays in the foreground, then the 3D chemical data must be compared to instrumental environmental records. This certainly also requires a robust temporal alignment of the shell record which daily growth pattern analysis can provide.*

This comment touches an important part of model development which may have been underrepresented in the current version of the manuscript. I thank Referee #1 for mentioning various strategies for testing the model results. Sub-annual growth patterns may be independently tested by comparison with the changing widths of daily growth increments. Care must be taken that this may lead to circular reasoning if these patterns are used as input into the model as suggested above. Additionally, an attempt shall be made to improve the model verification step by independently determining volume and the presence of microstructures through additional cross sections in longitudinal as well as in anterior-posterior direction. These additional sections will also allow chemical analysis on anterior and

posterior parts of the shells to verify heterogeneity in these directions and determine whether the model yields reliable estimates of shell chemistry away from the longitudinal cross section. I will attempt to incorporate these additional tests into a revised version of the manuscript.

*In conclusion, the paper needs rigorous rewriting and refocusing. Standard terminology must be adhered to. A native speaker must read the ms.*

*Selected other issues, more provided in annotated pdf:*

*In various places I missed proper references. For example, L65+66: "All these physiological changes, such as variations in growth and metabolic rate, shell mineralogy and spawning events, which affect the incorporation of trace elements into the shell of bivalves, complicate the use of trace element records to complement environmental reconstruction by stable isotope sclerochronology, (Klein et al., 1996b; Gillikin et al., 2005; Immenhauser et al., 2005; Freitas et al., 2006)." None of the cited articles refer to spawning-controlled element load.*

In a revised version, I will go through the references and revise them accordingly. More basis will be added for the influence of the reproductive cycle on trace element incorporation in bivalves.

*L77: Paper "under review" cannot be cited.*

The paper is currently in press and will be published in due time. I will upload its reference as soon as it is available online.

*L116-118: "Errors of reproducibility of μXRF measurements are generally higher than the instrumental error and depend on the integration time and the excitation energy of the element (see de Winter and Claeys, 2017; de Winter et al., 2017)." Is there really no reference predating 2017 that arrived at the same conclusion?*

I will try to cite earlier work on this phenomenon. However, the use of μXRF analysis is relatively underrepresented in geological literature.

*L122-123: "Timing of shell deposition was inferred from annual cyclicity in trace element profiles." [cyclicality!] No reference provided for this assertion.*

This touches on the issue raised above, namely that timing of shell growth needs to be better constrained in this study. This will be taken care of in a revised version.

*L95/96: Shells were "sectioned longitudinally along their axis of maximum growth using a slow rotating, diamond coated saw".*

It is not clear to me what Referee #1 means by this comment

*Reference list is done sloppy. Genus and species must be italicized. In "_18O" 18 must be superscript. Nouns in journal names must be capitalized.*

More attention will be given to the formatting of the reference list in a revised version.

*Figure 2: Meaning of colors in XRF scan?*

The meaning of the colours in the XRF scan are given in the figure caption. However, for clarity, a legend will be added to the figure.

*Figure 4: Size of letters varies too much (scales and axes' numbers). Why are annual growth lines not contouring the microstructure (compare e.g., Mouchi et al doi:10.1007/s00227-016-3040-6). Why are some years in enlarged hinge image flipped by 180_? Use "μ" not "u". Scale bars are far too thick. X-axis of graph must be mirrored (most recent year must be to the right, time axis should have same orientation as shell in topmost figure. Error bars missing.*

This figure will be edited according to the comments by Referee #1. The position of annual growth lines will be re-evaluated in comparison with previous literature (e.g. Mouchi et al. and Barbin et al.) and may be revised if needed.

*Figure 6: Purpose of plotting 2D shell again? You are showing 3D chemical data. . . Numbers on axes are far too small. Green is not legible. Do not use colors for axes' labels. Where are the error bars?*

Based on this comment and a similar one from Referee #2, the 2D shell scan will be removed from Figure 6. Axis labels and error bars of this figure will be revised.

*Please also note the supplement to this comment:*

*https://www.geosci-model-dev-discuss.net/gmd-2017-137/gmd-2017-137-RC1- supplement.pdf*

*Interactive comment on Geosci. Model Dev. Discuss., https://doi.org/10.5194/gmd-2017-137, 2017.*

***Comments posted as annotations in supplement:***

***Page: 1***

*For which purpose is a growth model really useful? Author did not provide justification. Author has not provided detailed description how he computed "daily" / "sub-annual" growth rates from annual increments. I am not aware of any bivalve that grows uninterruptedly through lifetime. One would not see changes in microstructure and no growth lines if growth proceeded with the same rate at all times. I can obtain information on growth rate much more reliably by analyzing daily etc. growth patterns. Why is the total daily/annual element load of a valve (only one valve was analyzed, not the entire shell) relevant? We know well that different microstructures differ chemically, even those formed at the same time. None of respective papers were cited. If the microstructure changes in different contemporaneous shell portions, then the chemical differences in shell portions reflecting the same amount of time is largely controlled by the microstructure. This leads to the most relevant question: What is this model good for? Leaving aside for a moment that we do not really know yet if the trace element chemistry provides robust environmental proxies, if I was to analyze the chemical variations, I would stick to the same microstructure and develop individual transfer functions for each microstructure and stick to line scans. If the author thinks that the new method is advantageous, then he should provide evidence with actual environmental data.*

As stated above, this comment is largely a result of the lack of clear statement of the aim of the model in the Introduction. I will rewrite this part of the manuscript in order to make this clearer. The interpolation in the model may be improved by using independent measurements of daily growth increments or a sinusoidal seasonality model into the model. This will improve sub-annual reconstructions made by the

model. As stated above, the aim of this model is not to prove that different microstructures have a characteristic chemical composition. Rather, the aim is to model total shell trace element incorporation rates to provide additional information about the uptake of trace elements from the environment that cannot be obtained by limiting analysis to one microstructure.

*Page: 2*

*of the ambient*

This will be rephrased

*odd phrasing! physiological changes comprise more than growth*

This will be rephrased

*but also as archives of environmental conditions (pollution etc.) and ecosystem change*

Indeed, I will add this

*odd phrasing! "physiological changes ... overprint chemical tracers" why not 'vital effects overprint chemical proxies of environmental change'?*

This will be rephrased

*This implies that there are models capable of modeling shell growth based on physiology. Cite those!*

This is not the case, I will rephrase this to better convey the meaning of the phrase. This should read: "without a priori knowledge of the pattern of morphological development of the species". This is important, because it means that this model could in theory be applied on fossil shells if they are well-preserved.

*What do you mean with "development"? Morpohological shape of the shells? Rephrase*

This will be rephrased to "growth and morphological development of bivalve shells"

*This is impossible based on annual increments, because shell growth varies on a seasonally time scale due to multiple different factors. You did not show any convincing evidence that your model performs well. This could be accomplished by comparing it to daily increment measurements.*

Indeed, this was mentioned above, and such an incorporation of seasonal growth rate changes into the model will be attempted in a new version.

*So what is the advantage of your model over the sclerochronological approach, i.e. counting growth increments and measuring their width?*

As mentioned above, the advantage is the possibility to reconstruct volume and mass changes as well as uptake rates of trace elements. I will try to state this more clearly.

*This method already exists since more than 100 years: growth increment count and width measurements*

This will be rephrased to something in the line of: "paving the way for mass and volume growth rate estimations"

*too imprecise, low detection limits*

Indeed, however the phase analysis applied on this model makes use of the statistics of larger groups of pixels in the XRF map to solve this issue. Precision of a typical phase area that contains tens to hundreds of thousands of pixels is better than that of a single pixel.

*How do you know that the chemistry and microstructure stays the same in the anterior and posterior portion of the shells, i.e. in any other sections other than the cross-section in which you have done measurements? This is not the case as demonstrated by Carriker et al (1991 Mar Biol 109, 287-297). Besides that, in different cross-sections, the arrangement of growth increments and relative proportion of the different microstructures can vary (and with it the chemistry!). Your model is based on a single cross-section. A three-dimensional view of the shell is needed, not a model. And again, if you claim that your model closely approximates the reality, then provide evidence, i.e. by studying the shell chemistry etc. with an independent method.*

A valid point of criticism, and one that can only be solved in part, as stated above. The presented model provides an approximation of reality, based on good measurement statistics on the 2D cross section. However, in a revised version, heterogeneity in shell chemistry will be tested more rigorously, including in the anterior-posterior dimension of the shell. This will provide independent checks on the performance of the model.

*But this has not been shown, though your phrasing implies you did.*

The model does approximates these incorporation rates, however more independent testing is needed to prove that the estimations of shell chemistry are representative of the actual shell heterogeneity in three dimensions (see above).

*Only done for one (2D) transect, not 3D as model!*

See above, more rigorous testing will be executed in a revised version of the manuscript.

*Mineralogy? You said it is calcite, so only one mineral. The different textures are crystallographical differences.*

This terminology is indeed confusing, and I will refer to these textures as "microstructures" in a revision.

*Not measured. Growth rate is not necessarily mineralization rate.*

This will be rephrased

*Phrasing implies you tested this and actually showed that this gives a better insight.*

This will be stated more carefully in a revised version

*We are far away from routinely applying trace elements of bivalves as environmental proxies, because there is vital and perhaps kinetic effects which your method cannot model.*

Indeed, however the aim of this model is to bring us one step closer to the application of these proxies by considering the total shell trace element uptake rates, which provide us with additional information about the fluxes of trace elements in bivalves next to the more conventional line scan measurements.

*Page: 3*

*I would not consider a speleothem or most bivalve species good examples of fast-growing carbonates. What these (climate) archives unites is periodic growth.*

This will be rephrased. What is meant is that these archives allow proxy records of high temporal resolution to be made.

*A bit odd to see these citations here. Cite pioneering studies!*

This is a valid comment and better references will be added.

*There is not just this polymorph in mollusks, but mainly aragonite, occasionally vaterite and ACC.*

Indeed, but the fossilization potential of all but the calcite is rather poor. I will try to rephrase to make clear that I refer to the fossilization potential and do not mean that this is the only polymorph in a bivalve shell.

*and many more studies have demonstrated that the element chemistry is difficult to interpret. Since this is in the focus here, it needs to be reported!*

I agree and will add this to the revised text.

*odd phrasing, specimens can be abundant, and the number of species high (species richness)*

This will be rephrased

*Not true for stable carbon isotopes. There are also bivalves in freshwater, which supposedly form their shells in oxygen isotopic equilibrium with the ambient water as well.*

This will be rephrased to make clear that bivalves are not strictly marine and that the focus is on oxygen isotopes here. The focus of this manuscript is on marine bivalves, although there is no fundamental reason why the model would not be applicable on freshwater shells.

*also on a subseason, daily, tidal, fortnightly time scale. --> sub-annual time scale!*

This will be rephrased

*Unclear. You need to better describe what the limitations of d18O values are. And leave out d13C, because this system is not well understood.*

This will be rephrased to lay the focus on oxygen isotopes. I will shortly mention carbon isotopes separately in a revised version of the manuscript.

*carbonate !*

This will be rephrased

*... to obtain independent measures of environmental variables. Assumption is that some trace elements are indicative of changes of only a single environmental variable (unlike d18O which record changes of T and S at the same time).*

Correct, I will add this for clarity as it is indeed important.

*And at least double as many studies have shown that it is very challenging to do so and requires, for example, a mapping of the microstructures etc.*

This will be added to the introduction, as mentioned above

*since when?*

This will be rephrased

*I do not understand how this paragraphs relates to the last. Transition!*

The paragraph serves to introduce some earlier work on oyster shells. I will try to make the transition better.

*What kind of "development"? Do you mean ontogenetic development of the morphology of the shell? Why not simply say: ontogenetic changes of the shell (trace and minor element) chemistry?*

This will be rephrased

*What has this to do with chemistry?*

I will reformulate this to make the link clear. This will be a good moment to introduce the citations for the different chemical signatures of these microstructures, which are suggested by Referee #1 in earlier comments.

*shell microstructures*

This will be rephrased

*they are!*

*associated with*

This will be rephrased

*Page: 4*

*I am not aware of any study that used the chemical properties of the entire shell portion formed in a specified time interval to reconstruct environmental variables. However, it is an interesting approach! Whether it is feasible to measure and needed is another question. Reasons I can think of why changes of the volume of shell precipitated in a given time interval matter are to determine changes in ecosystem productivity or CO2 sequestration by bivalves.*

This is indeed the novelty of the model presented in this manuscript. I will try to state this more clearly in the Introduction of the manuscript. In my opinion, the reconstruction of total shell chemical composition and its changes through time have uses outside the estimation of ecosystem productivity and CO2 sequestration. As mentioned above, these reconstructions may also shed light on trace element fluxes into the bivalve shell and may give more information about what drives the uptake rate of these trace elements.

*This implies that the changes in texture, chemistry etc. are purely driven by physiology of the bivalve, which is not true. Rephrase.*

This will be rephrased

*??? By the way, non of the cited articles mentions 'spawning' as a cause of trace element incorporation.*

These references will be updated (see above)

*delete comma*

*and between specimens (see Gillikin et al)*

This will be added, as it is indeed an important observation

*were*

*You cannot cite paper under review. And in addition, there are most likely citable published papers. Your model is likely not capable of doing a better job than to apply the actualism principle.*

The citation will be updated as soon as the paper is published. The goal of the model is to provide information on these development curves that can then be compared with modern growth curves. This yields independent control on the shell development which could prove useful in fossil bivalve studies.

*New paragraph*

*But then I do not understand how can one arrive at monthly or daily resolution. That's impossible to model without knowing the seasonal rate of growth and possible changes of the duration of the growing season and rate through lifetime. We already have tools to determine that: daily increments (see Barbin et al).*

A revised version of the manuscript will include a revision of the model in which these considerations are taken into account. Indeed, the incorporation of seasonal changes in growth rate and growth stops will make the model more reliable.

*?*

This will be rephrased to "growth pattern"

*One could be picky here: You cannot model the incorporation, but only what is left in shell (early diagenesis!). Rephrase: model the ontogenetic changes of trace elements in the shell. But what for???*

This will be rephrased, the aim of the model will be stressed more clearly to avoid confusion about the goals of modelling total shell trace element concentrations and accumulation rates.

*A first-time application must be verified with independent methods.*

Measurements of daily growth increments may serve as independent verification.

*2D vs 3D comparison not valid. Previous studies... different species, different specimens?*

More effort will be done to check model result in a revised version of the manuscript (see above), but in my opinion comparison of the obtained growth patterns with growth curves obtained in previous studies should be retained as a check on the model results.

*reason?*

This step was carried out to disinfect the shells and to remove algae from the outside of the shell.

*del s*

*-*

*Are you sure you did not measure shell height (umbo - ventral margin)? At least the cross-section in Fig. 2 etc. is through the height axis.*

Indeed, shell height was measured. The terminology of all shell dimensions will be corrected in a revised version.

*I assume the saw was not coated, but the saw blade!*

Correct, this will be rephrased

*grit*

***Page: 5***

*Number*

This will be rephrased

*Furthermore, q...*

This will be rephrased

*Unciteable*

This reference will be updated

*Nobody else has done this before?*

I will try to find an earlier reference for thin in μXRF, though the technique is not very common (yet).

*huge for some elements...*

Relative standard deviations of reproducibility can be high (especially for Zn), but chemical differences between microstructures are statistically significant.

*you have only used one method, no need to repeat here*

Phrase will be removed.

*Unclear to me. You are making certain assumptions here that need justification, e.g. reference to previous work on seasonal element cycles. Do you mean seasonal timing as well?*

Agreed, as mentioned above, the age model needs more independent constraint. This will be attempted in a revision.

*Stick to standard terminology in the field! Increments typically refer to the shell portion between two growth lines/bands. Of course you could also stick to Douglas S. Jones's terminology and refer to the*

*growth lines as 'increment I' and to growth increments as 'increment II', because in some species the growth lines are not distinct , but merely bands.*

The terminology throughout the manuscript will be revised to avoid confusion.

*refer to figure, explain better*

An attempt will be made to better explain this procedure

*Page: 6*

*shell thickness increases as the bivalve add shell material to the inner shell surface*

This will be rephrased

*Height*

Terminology will be updated

*Page: 7*

*My recommendation is that you callaborate with a bivalve sclerochronologists.*

*You are introducing new terms without explaining them. What is a "sub-increment" and how much time is represented by it? How do you legitimize a sub-increment? How do you know when they formed? Again, you likely mean sub-line and not increment (see comment above).*

See above, indeed "sub-line" will be better.

*Doesn't this assume that the shell thickness is unchanged at any shell portion? No real shell is like that.*

It does not, shell thickness tapers down towards the anterior and posterior side of the shell. This is illustrated in Figure 3C. The length (in anterior-posterior direction) to height (umbo to outward edge) ratio of the shell is kept constant, meaning that length (Z-direction) changes as height changes for every sub-increment.

*Page: 8*

*Again, for an idealized shell that may apply, but not for a real shell.*

The modelling of the shell in Z-direction is indeed an approximation

*Page: 9*

*How do you know that shell density remains unchanged through lifetime of the animal? Most certainly incorrect, because the shell microstructure (and thus density) changes with age and the relative proportion of the organic versus the carbonate phase. The 1 to 5 wt% organics is just an average value.*

The model might indeed improve by incorporating variable density based on density measurements of microstructures coupled to the XRF map phases. An attempt will be made to incorporate this into the model in a revised version.

*Page: 10*

*Term is occupied to distance between apices of valves.*

Terminology will be adapted, as the anterior-posterior distance is meant here

*Are you aware of the possibility to determine the ontogenetic age by counting annual growth increments?*

I am, but this maybe complicated by the fact that changes in microstructure are not always annual (Surge et al., 2001). Nevertheless, a revision will include better age constraint and an attempt to use daily growth increments to constrain shell age.

*Is your goal to compute max size, weight etc of individual specimens?*

This is not a major goal of the manuscript

*Nacre is aragonite! As a bioceramic it just has a specific microstructure that is not present in the abiogenic aragonite crystal.*

This will be rephrased and the proper terminology will be used.

*Page: 11*

*On which basis have you distinguished these four element levels? Are the data sets statistically significantly different? Then, please show statistics.*

The separation is made by PCA assisted phase analysis of the XRF maps and the differences between phases are statistically significant (see also Table 3).

*at the outer shell surface*

This will be rephrased

*Phrase better! Shell portions with foliated and chalky calcite are chemically different.*

This will be rephrased

*But not exclusively, right?*

No, I will rephrase and refer to Figure 4 for clarity

*I am not sure exactly what you mean. Addition of shell material in bivalves occurs in two separate compartments, extrapallial spaces (EPS). In the the inner EPS, the inner shell layer form, which contributes to the thickening of the shell. In the outer EPS, the outer shell layer forms which mainly ensures growth in size (height and length), until later mature stages of life when the outer shell layer grows inward further contributing to shell thickening at the ventral margin.*

I thank Referee #1 for this comment. I will attempt to rephrase this sentence to clarify this in the manuscript text.

*You explain well-known things in difficult ways and without using proper terminology.*

This will be rephrased to clarify

*Other species grow in a similar manner but may have serves as an easier start for this approach.*

Indeed, but the aim of this manuscript is to test the model on irregular oyster shells to demonstrate that it works on morphologically heterogeneous shells.

*Justification? Periodic means time. I assume this has been studied previously, so refer to such papers.*

I will add references and also improve independent time constraint in a revised version of the manuscript.

*Rephrase. A crystal cannot be diluted with material.*

This will be rephrased

*I assume, growth lines are here referred to.*

Indeed, terminology will be updated in revision.

*see comment above on "periodicity"*

See reply to that comment

*age model??? There are annual and daily increments that are perfectly suitable for this purpose.*

Good point, an analysis of these will feature in a revised version.

*Based on literature data it appears that the different textures are not formed simultaneously but after another (e.g. Mouchi et al doi:10.1007/ s00227-016-3040-6)*

I will try to provide suitable references for this

*increments,*

This will be rephrased

*"paced to the seasonal cycle": I do not understand what you mean.*

This will be rephrased. I meant to say that the location of the increments does not correspond to cyclicity in the trace element records. This may, however, be a result of the location of the line scan (as pointed out by Referee #2). In a revised version, I will add more line scan measurements to illustrate the effect of incorporating different microstructures and measuring on different locations on the shell cross section to verify the model results.

*Are you claiming that different textures are formed at the same time? Please provide reference that support this assertion. Your data are not sufficient to support this claim. As far as I can tell from the figure, the shell textures and growth patterns coincide, i.e. foliated and chalky textures alternate through time, not space.*

I will try to provide suitable references for this.

*Of course, growth lines are formed simultaneously in different shell portions. What do you mean?*

I will remove "isochronous" as it is indeed a pleonasm in this sentence.

*again provide references*

I will try to provide suitable references for this or rephrase the discussion in this paragraph.

*These are just different microstructures, but not different mineral phases.*

Terminology will be updated.

*Rephrase. This is not English.*

This will be rephrased

*Ok, we are coming closer to what your intention might have been: You are studying the change of the average daily chemistry of the shell. However, if your claim is correct that the element concentration is linked to microstructure (which has been reported many times before for other species!) and different microstructures are formed at the same time (which is also nothing new), then what is the advantage of determining how the average daily, monthly or annual chemical composition of the whole shell has changed?*

Please refer to the explanation of the aim of the model in earlier comments. I am a bit confused by the fact that Referee #1 mentions that it is known that microstructures are formed at the same time ("which is also nothing new"). A few comments earlier I thought that Referee #1 mentioned that Mouchi et al. demonstrated the opposite. Maybe I misunderstand this comment. In any case, I will sort out the literature for this question and adapt the discussion to incorporate the right references.

*Page: 12*

*I do not understand the rationale for needing a "growth model". Can you not just read the growth patterns in the shell? So far, I have not understood the impetus of the whole paper. Is the author familiar with Barbin et al 2008 (DOI 10.1007/s00531-006-0160-0)? There are daily increments in these bivalves that provide a much easier - and more robust - means to measure time*

I hope that a revision of the Introduction and Discussion of the manuscript will clarify the aim of this modelling approach (see above). An attempt will be made to include measurement of daily increments into the manuscript as independent age control.

*So what? Can't we get the same result by measuring the dimensions of shells of different age/size classes of oysters? How have you computed the rate of growth in each month?*

This just proves that modelled growth curves give similar results than those reported in literature. I hope to demonstrate in a revised version that constructing growth curves is not the only aim of the model, but it might be an important application in the study of fossil bivalves which we cannot culture and measure.

*Page: 13*

*g,*

*Right, model makes incorrect assumptions, 2D to 3D interpolation is an oversimplification.*

Indeed, some of this offset may be explained by oversimplification. However, this offset in mass and volume is not too bad in my opinion, considering the irregularity of oyster shells. It will most likely be much more accurate in more regular bivalves.

*I guess you mean anterior and posterior ends*

Indeed, terminology will be adapted in a revised version.

*I do not understand where the information on seasonal (or even daily) shell growth is derived from.*

It is interpolated, but incorporation of seasonal growth variations in the model would be a good improvement indeed (see above).

*temperature is relevant as well (see e.g. work by Ansell 1968).*

This is a good comment, I will add this reference to the discussion.

**Page: 14**

*??? I do not know what this means here.*

This will be rephrased to "morphological development"

*How do you know that growth rate influences the shell chemistry? It could be environment, ageing, physiological control etc.*

Accumulation rates of trace elements in the shell correlate with growth rate, and these changes are forces by the relative proportion of different microstructures in the newly formed parts of the shell. Therefore, there seems to be a control of shell growth rate on microstructure and chemistry. The point I am trying to make is that these types of comparisons can only be made when total shell concentrations are considered, demonstrating the added value of the model.

*Where is the evidence? Growth patterns are just enough. As far as I understood you claim is that textures vary in different contemporaneous shell portions and some elements covary with the microstructure. How can then the chemistry help to double-check growth patterns/rate?*

This section will be rephrased in a revised version of the manuscript. What I meant to say is that the model allows growth rates (in terms of volume and mass) to be reconstructed as well as trace element accumulation rates, meaning that the influence of changing growth rates on the incorporation of trace elements into the shell can be studied.

*I do not understand how your data can provide information on seasonal growth rates if you just measured annual increments. The seasonal growth rate is not just controlled by the physiological ageing but also food supply and temperature etc. Have you taken these effects into consideration in your model?*

This is a good comment and application of seasonal changes in growth rate will help improve the model results on the sub-annual timescale.

*I do not understand how this is possible. Show please with actual environmental data that your model can provide that information.*

This needs to be rephrased, as I agree that I cannot compare my model results with actual environmental data. Such comparison is also outside the scope of this manuscript. What is meant is that the influence of changing growth rates on trace element uptake can be reconstructed using this model. In theory, this allows the elimination of the effects of growth rate changes on trace element records and a discussion of total shell uptake rates of trace elements from the environment.

*Again, if there is a change in microstructure, then there is change in chemistry. If there is a greater proportion of foliated layer, then your "total ... element content" of that year is higher/lower than in another year with a different ration of foliated versus chalky shell. If sclerochronologists use trace elements, they are well aware of the fact that analyses should be completed within the same texture!*

I agree, and the additional information about changes in microstructures that are formed which is given by the presented model may be used to constrain total shell uptake rates of trace elements, which in turn may give information about ambient trace element concentrations additional to the concentrations of a single microstructure.

*Page: 15*

*capitalize nouns*

This will be changed

*why capital letters?*

This will be changed

*Page: 16*

*Italicize*

This will be changed

*superscript numbers*

This will be changed

*superscript numbers*

This will be changed

*italicize genus, species and subspecies!*

This will be changed

*Page: 23*

*Where are the annual growth lines?*

Annual growth lines are indicated in yellow and white

*scale bars to thick!*

This will be changed

*wt%*

This will be changed

*what is "ug". Use lower case Greek "μ", please*

This will be changed

*why are some years 180° flipped? Place all years in growth increments, not on top of growth lines.*

This will be changed, it was meant to make the dates more legible

*mirror plot: time to the right as in topmost figure*

This will be changed

**Reply to comments raised by Anonymous Referee #2**

Dear Editor and Referees,

Please find below a point-by-point reply to the comments and remarks posted by Referee #2 on my manuscript titled "ShellTrace v1.0 – A new approach for modelling growth and trace element uptake in marine bivalve shells: Model verification on pacific oyster shells (*Crassostrea gigas*)", which was submitted for interactive discussion to Geoscientific Model Development (gmd-2017-137)

*Review of de Winter "ShellTrace v1.0 - A new approach for modelling growth and 1 trace element uptake in marine bivalve shells: Model verification on pacific oyster shells (Crassostrea gigas)"*

*This manuscript evaluates the trace elemental composition of the shells of the pacific oyster using an integrated proxy and modelling approach. There is currently significant debate in the peer-reviewed literature regarding the use of trace elemental ratios derived from marine bivalve molluscs as palaeoenvironmental proxies. This debate is largely associated to developing an understanding of the contribution of environmental conditions and vital effects have on the trace elemental composition. These effects have been reported to create significant heterogeneity in the trace elemental composition of bivalve calcium carbonate and thus developing a robust quantitative understanding of these effects is required if we are to reach a stage where trace elemental ratios could be applied in a robust palaeoenvironmental application. Therefore a detailed study utilising modelling and actual shell data is very interesting.*

*This study takes an interesting approach to developing an understanding of the relationships between shell growth metrics, age and trace elemental composition. I think that approaches undertaken in this manuscript could have the potential to significantly develop our understanding of the drivers of trace elemental composition in oyster shells and potentially other long-lived marine bivalves, if more widely applied. However, as the manuscript is currently presented there appear to be significant flaws in the methodology that lead to the stated conclusions being unsupported by the presented data. I have serious misgivings about the assumptions that appear to have been applied in the development of the model and in the manner in which the data are presented. The manuscript therefore needs significant revisions before it reaches a publishable standard.*

*General concerns:*

*My main concern revolves around the generation and interpretation of trace element data, and the lack of an independent age model based on the interpretation of growth increment patterns. Firstly, the line used for sampling the trace elements, as shown in Figure 4, does not follow the axis of maximum growth (which is the conventional approach for sampling geochemistry in shells) and is not perpendicular to any of the growth increments sampled. This means that the temporal averaging of the trace elemental ratios in each analyses in each increment is inconsistent. This makes it very difficult to compare data between*

*increments. Other complications such as the potential of non-linear seasonal growth rates are not considered. It is not clear from the data that are presented that there are any consistent seasonal patterns in the elemental ratios. The author must provide a statistical assessment of the mean seasonal trend for each element analysed. As this is not clear it is near impossible to evaluate if the relationships between elemental ratios and age is robust. I go through each of these points in more detail below.*

Referee #2 raises some valid structural problems with my work, most of which are also put forward by Referee #1. One major concern that both Referees raise is the way in which an age model is constructed for the shell records. I acknowledge that this is a weak point in the paper and that it should be improved in a revised version. One way to improve the age model is put forward by Referee #1, namely to use independent chronology from the measurements of daily growth increments to date XRF line scan data. I will attempt to implement this method in a revised version of the paper.

Another valid criticism of Referee #2 are the locations of XRF lines, which are not perpendicular to the growth lines. The reason why these locations were chosen is that I attempted to measure trace element profiles through the foliated calcite in the shell hinge and to avoid sampling of the chalky calcite microstructure. This sampling strategy allowed the comparison of purely foliated calcite trace element profiles with the total shell concentration records given by the model. Such sampling could only be achieved by measuring a line close to the shell hinge, where the chalky calcite is almost entirely absent. However, I realize that sampling at an angle with respect to the axis of maximum growth creates its own problems with respect to temporal averaging (see comment above). I therefore propose to apply a more rigorous sampling approach for XRF line scans in a revised version of the manuscript, in which I compare model results to both records through the foliated calcite and records through the axis of maximum growth (which include chalky calcite layers). In the revised manuscript, a more comprehensive discussion of the differences in result between these sampling strategies will be included to highlight the advantage of records generated with the proposed model. In this discussion, a more rigorous testing of the presence of seasonal trends in the data will also be included.

*Detailed comments:*

*Line 38-39 – This is a strange list of references to use. There are many more papers that arecurrently already published that would be better to cite here. For example (Butler et al., 2013, Butler et al., 2011, Reynolds et al., 2016, Wanamaker et al., 2008a, Wanamaker et al., 2008b, Wanamaker et al., 2011, Schone et al., 2003, Schöne et al., 2011, Jones, 1980, Jones et al., 1989, Witbaard et al., 2005, Witbaard et al., 1997, Witbaard et al., 2003, Witbaard et al., 1994, Swart et al., 2010)*

I thank Referee #2 for providing me with suggestions to improve the literature review in the Introduction, and will consider using the suggested papers in a revised version of the manuscript.

*Lines 41-42 – The author needs to cite a better range of literature.*

See comment above and comments from Referee #1: I will restructure the Introduction in a revision of the manuscript to include a more comprehensive literature basis to introduce the subject.

*Line 47 – should not cite work that is not published (remove deWinter et al., PPP)*

This paper is currently in press and I will update the reference as soon as it is available online.

*Line 39 – not all bivalves are calcite, change to calcium carbonate. This term bivalve calcite is used throughout the manuscript and should be changed.*

I agree with comments from both referees that the terminology used in the manuscript needs a thorough revision and will implement suggestions from both referees to improve this in a revised version.

*Lines 50-52 it would be useful for the author to expand on this statement to say exactly which trace elements have been used to examine which environmental archives and importantly from which bivalve species these studies were conducted. MY experience is that there is a lot of contradictory literature surrounding the application of trace elements as environmental proxies and so a far more detailed discussion around this is needed.*

I acknowledge the criticism raised by Referee #2 regarding the literature surrounding the application of trace element proxies in bivalve shells, and a similar point was raised by Referee #1. In a revised version, I will incorporate a more comprehensive review of the literature about the use of trace element concentrations in bivalve shells, also focusing more on the difficulties in using trace elements as proxies in these records.

*Lines 53-59 – Again the author needs to provide a far more detailed discussion of what these previous studies into oyster shells found. "There is some discussion about the role of these calcite mineral phases, whether their precipitation is controlled by environmental conditions and whether changes in the precipitated mineral phase are paced to regular (solar or lunar) cycles" So what do they actually say in this discussion? What environmental factors do they suggest are responsible? Were there any issues identified in these studies? The author needs to expand significantly here.*

See above, a revision of the manuscript will include a more comprehensive literature review.

*69-71 – The author needs to be more concise in the structuring of sentences throughout the manuscript. As it is currently written it is quite difficult to read or. One example for sentences that could be reduced is: "Furthermore, several studies have shown that rates by which trace elements are incorporated into bivalve shells and the degree to which these rates are controlled by environmental factors can be vastly different between different bivalve species". This could be condensed to, for example, "Studies have shown that the rates of incorporation, and environmental response of, trace elements in bivalve shells varies between species"*

I thank Referee #2 for suggestions on how to improve the legibility of the manuscript and will attempt to simplify the structure of my sentences in a revision.

*77-85 – this section reads far too much like a methods summary and not a set of aims or hypotheses that this study aims to test. This should be rephrased.*

This comment is similar to criticism raised by Referee #1 about the Introduction lacking a clear statement of the aim of the study. The Introduction will be rephrased during manuscript revision to clarify the aim of the modelling approach and give the manuscript a better structure.

*88 – I presume not, but please state whether the shells cooked by the restaurant before they were analysed in this study? If so would this have had an impact on the elemental composition of the shell?*

Oyster shell were indeed left untreated by the restaurant. A mention of this will be added to the revised manuscript.

*90 - from what water depth were the shells collected?*

A revised version of the manuscript will include a statement about the water depth in which the shells were grown, which was between 5 and 10 meters.

*105 – clarify that you mean the polished shell surface.*

This will be rephrased in the revised version.

*112 – "using the point-by-point scanning method outlined in de Winter et al. (in review, PPP)" Firstly you shouldn't cite work that is not published. Secondly we cannot see what this method is as it hasn't been published. We therefore cannot assess if this is a robust methodology. The author should elaborate on the details of this methodology.*

See above, the cited paper is currently in press and its reference will be updates as soon as it is publicly available.

*281 – why would you estimate age based on proxy records? Growth increment analyses are a far more robust measure of ontogeny.*

This is a valid comment and one of the major criticisms by both referees and will be addressed in detail in a revised version (see above and in reply to Referee #1).

*309-311 - "Shell increments used as tracers for growth modelling are generally characterized by decreased Ca and Mg concentrations and increased concentrations of Fe, Mn, Zn and Sr." This is not obvious from Fig 4 at all. I personally cannot see this interpretation in these data the way they are plotted. If there is a consistent response in the elemental ratios throughout the year, with either peaks or troughs corresponding to the growth lines then the authors should generate a mean seasonal elemental curve using these data and plot with error bars the mean seasonal change in each element. This will then provide a more rigorous assessment of whether the ratios do indeed show a change occurring around the timing of the growth check forming. However, I am concerned by the trajectory that was used to generate these data as shown in Figure 4 (line A-B). The XRF sampling line (line A-B) does not follow the axis of maximum growth through the hinge region of the shell (the maximum growth appears to follow the position of the numbers 0-7). The position of the sampling line as is not perpendicular to any of the growth increments meaning the temporal positioning of each set of analyses through these increments is not consistent. This will lead to large changes in the temporal smoothing of the data through the timeseries and lead to significant challenges in interpreting the data. For instance this means that comparing one year to the next is very difficult as they incorporate a different temporal smoothing. The plotting of these data against a constant time axis is therefore inappropriate. Plotting the data like this makes the assumption that the growth of these shells is constant throughout the growing season and that the growing season is 12 months. Both of these assumptions are likely false as has been found in many other species of bivalve. The author needs to provide a far more robust approach to determining the timing and rate of seasonal growth before these data can be plotted against a time axis.*

This point will be addressed by adding independent age control using a study of the daily growth increments. This will increase the confidence of the age model applied on XRF trace element profiles in

this study as well as show how long the growth season was and how growth rates varied with the seasons. This information will help with the interpretation of trace element profiles and may also be used to improve the growth model itself (as suggested by Referee #1). Furthermore, as mentioned above, a more careful discussion of the locations of measurement applied in this study and the comparison with transects through the axis of maximum growth will be added. This will allow better control on the timing and rate of deposition of shell material by the oysters and improve the discussion of the trace element profiles and model results.

*Fig 4: This figure is extremely complicated to understand. The impression this figure gives me is that the sampling strategy of this study is fundamentally flawed. For example, the black lines, which apparently correspond to the growth increments show there are 7 increments yet these do not appear to line up with annual chronology in any way? Why is this? Also the panels on this figure need to be labelled and referenced more clearly in the fig legend.*

In the revised version an effort will be made to simplify this figure and improve the sampling strategy.

*312-313 – this needs to be tested.*

Better statistics on the seasonality in trace element profiles as well as an improved measurement strategy will demonstrate how trace element concentrations in these oysters vary through the seasons.

*313-314 – why did the authors use the elemental ratios to determine an age model when the growth increments would have provided a far more robust, and independent age model? Based on the data that is displayed in Figure 4 it is extremely difficult to see any consistent patterns that would facilitate the attribution of an age model with any degree of confidence. As mentioned above, if the elemental ratios are to be used as an age model, the data from each increment should be averaged to generate a mean seasonal curve for each element. These data, with associated error bars could potentially be used to assign ages, providing the data demonstrate a significant seasonal pattern.*

I agree with this critique, and will revise the measurement strategy of XRF lines and attempt to add independent age control by applying measurements of daily growth increments in the shells.

*315+317 – "Note that line scanning through the hinge of the shell only allows for the sampling of the last three growth years, as the irregular shape of the oyster shell and the occurrence of chalky calcite further up the hinge prevents the measurement of a complete record through the foliated calcite" The author needs to explain why this line was used for sampling and why the line that follows the position of the numbers in Figure 4, lower left panel, which contains 7 growth increments wasn't used. Using this line would have facilitated sampling through 7 increments.*

As mentioned above, this strategy was implemented to avoid sampling of chalky calcite. However, I realize that the discussion would be incomplete without also adding line scans through the axis of maximum shell growth and these will be added to a revised version of the manuscript.

*326 – Growth model section – I really don't understand why the author has not used the visible growth increments to develop the age model and instead used the trace elemental records. I am not convinced that the geochemical signatures provide a robust signal for age model development based on the data that has been presented in this manuscript. The author needs to provide significant new data comparing*

*growth increment based age models in relation to the geochemical records before these analyses can be assessed in a robust manor.*

See comments above: I agree and will attempt to improve the age constraints of the data presented in this study.

*344 – How does the Von Vertalanffy R2 statistic (0.60) compare to that of other bivalves?*

A revised version of the manuscript will include a comparison of this statistics with other Von Bertalanffy fits in the literature.

*391- Trace element model section. In this section the author presents "Records of trace element accumulation rates and total shell trace element concentrations" and discusses these data based on Figure 6 which plots these data against a time axis over the period from 2014-2017. As mentioned with Figure 4, these analyses assume a constant seasonal growth rate and that growth occurs all year round. Yet no evidence or citations are provided to support this claim.*

Indeed this is not the case. In the revised version of the manuscript, more effort will be put into showing the length of the oyster's seasons of growth and growth rates using independent control and a more robust discussion of the sampling strategy.

*Figure 6 – remove the scan of the shell in the lower panel as it is not needed. Also all the panels should be labelled (A, B etc.)*

The figure will be adapted according to the comments by Referee #2.

*403-404 - "These differences in total shell concentrations and concentrations in the shell hinge illustrate the value of the proposed trace element modelling approach" To be honest I am not sure why this is important. In the application of sclerochronological trace elemental ratios we are looking to explain the patterns of variability that we can measure along a 1 or 2 dimensional growth axis in the shell. Whilst it is of interest to understand the reasons behind heterogeneity in trace elemental ratios across the sampling plane, we are rarely interested in total shell elemental composition. The big question that needs addressing with trace elemental work is to what degree is the variability in trace elemental ratios, sampled along the axis of maximum growth, due to vital effects and what proportion is due to changes in ambient water chemistry. The author does hint at this in lines 406-411 however, I feel that this has not been achieved by this study.*

I agree with Referee #2 that the main hurdle for the application of trace element profiles in sclerochronology is finding out how much of the variation in these records is driven by environmental factors as opposed to physiology of the animal. However, I do believe that the modelling of total shell trace element concentrations and accumulation rates can shed light on this question. A record of total shell trace element uptake contains information about how much of a certain trace element is stored in a bivalve shell at any given point in time, something which cannot be known from analyzing a 2D profile through the shell or limiting analysis to one microstructure. The aim of the model presented in this study is to reconstruct the flux of trace elements into the bivalve shell through its life time. If trace element concentrations are controlled by changes in the environment, this total shell uptake rate should relate to the environmental concentration in a more straightforward way than the concentration of trace elements in a certain part of the shell or in one of the microstructures. The reconstruction of these trace

element fluxes into the shell might allow for a better distinction between environmental and physiological parameters that influence trace element uptake.

*418-419 – I strongly disagree with this statement. There is no evidence that this is the case presented in this study and should be removed.*

In a revised version, better care will be taken to support the claim that the assumptions on which the model is based allow it to be used on a wide range of bivalve shells. I hope that a revision of the model according to suggestions by Referee #1, including a consideration of growth halts and seasonal variations in growth rate, will convince the referees that the assumptions on which the model is based are reasonable. Also, a more rigorous testing of the model results will be added to the revised manuscript in the hope that the output of the model may be more thoroughly verified.

*422-424 – This statement if wrong. The assumptions made in this study if applied to other marine bivalves such as those commonly used in sclerochronology (e.g. Arctica islandica) would likely not hold. This study assumes constant annual growth which is not the case for A. islandica.*

See comment above: I agree that the model as it is presented in the present version of the manuscript has shortcomings that can be solved with help of the suggestions of both referees. A revised version of the model and the manuscript will take these shortcomings into account in order to improve the growth and trace element model and prove that its output models real life conditions.

*Suggested references*

*Butler, P. G., A. D. Wanamaker, Jr., J. D. Scourse, C. A. Richardson and D. J. Reynolds (2011). Long-term stability of delta c-13 with respect to biological age in the aragonite shell of mature specimens of the bivalve mollusk arctica islandica. Palaeogeography Palaeoclimatology Palaeoecology, 302, 21-30.*

*Butler, P. G., A. D. Wanamaker, J. D. Scourse, C. A. Richardson and D. J. Reynolds (2013). Variability of marine climate on the north icelandic shelf in a 1357-year proxy archive based on growth increments in the bivalve arctica islandica. Palaeogeography, Palaeoclimatology, Palaeoecology, 373, 141-151.*

*Jones, D. S. (1980). Annual cycle of shell growth increment formation in 2 continental-shelf bivalves and its paleoecologic significance. Paleobiology, 6, 331- 340.*

*Jones, D. S., M. A. Arthur and D. J. Allard (1989). Sclerochronological records of temperature and growth from shells of mercenaria-mercenaria from narragansett bay, rhode-island. Marine Biology, 102, 225-234.*

*Reynolds, D. J., J. D. Scourse, P. R. Halloran, A. J. Nederbragt, A. D. Wanamaker, P. G. Butler, C. A. Richardson, J. Heinemeier, J. Eiriksson, K. L. Knudsen and I. R. Hall (2016). Annually resolved north atlantic marine climate over the last millennium. Nature Communications, 7.*

*Schone, B. R., W. Oschmann, J. Rossler, A. D. F. Castro, S. D. Houk, I. Kroncke, W. Dreyer, R. Janssen, H. Rumohr and E. Dunca (2003). North atlantic oscillation dynamics recorded in shells of a long-lived bivalve mollusk. Geology, 31, 1037-1040.*

*Schöne, B. R., A. D. Wanamaker, J. Fiebig, J. Thébault and K. Kreutz (2011). Annually resolved _13cshell chronologies of long-lived bivalve mollusks (arctica islandica) reveal oceanic carbon dynamics in the*

temperate north atlantic during recent centuries. Palaeogeography, Palaeoclimatology, Palaeoecology, 302, 31-42.

Swart, P. K., L. Greer, B. E. Rosenheim, C. S. Moses, A. J. Waite, A. Winter, R. E. Dodge and K. Helmle (2010). The c-13 suess effect in scleractinian corals mirror changes in the anthropogenic co2 inventory of the surface oceans. Geophysical Research Letters, 37.

Wanamaker, A. D., Jr., J. Heinemeier, J. D. Scourse, C. A. Richardson, P. G. Butler, J. Eiriksson and K. L. Knudsen (2008a). Very long-lived mollusks confirm 17th century ad tephra-based radiocarbon reservoir ages for north icelandic shelf waters. Radiocarbon, 50, 399-412.

Wanamaker, A. D., K. J. Kreutz, B. R. Schoene, Pettigrew.N., H.W. Borns, D. S. Introne, D. Belknap, K. A. Maasch and S. Feindel (2008b). Coupled north atlantic slope water forcing on gulf of maine temperatures over the past millennium. Climate Dynamics, 31, 183-194.

Wanamaker, A. D., K. J. Kreutz, B. R. Schöne and D. S. Introne (2011). Gulf of maine shells reveal changes in seawater temperature seasonality during the medieval climate anomaly and the little ice age. Palaeogeography, Palaeoclimatology, Palaeoecology, 302, 43-51.

Witbaard, R., G. C. A. Duineveld, T. Amaro and M. J. N. Bergman (2005). Growth trends in three bivalve species indicate climate forcing on the benthic ecosystem in the southeastern north sea. Climate Research, 30, 29-38.

Witbaard, R., G. C. A. Duineveld and P. Dewilde (1997). A long-term growth record derived from arctica islandica (mollusca, bivalvia) from the fladen ground (northern north sea). Journal of the Marine Biological Association of the United Kingdom, 77, 801-816.

Witbaard, R., E. Jansma and U. Sass Klaassen (2003). Copepods link quahog growth to climate. Journal of Sea Research, 50, 77-83.

Witbaard, R., M. I. Jenness, K. Vanderborg and G. Ganssen (1994). Verification of annual growth increments in arctica islandica l from the north-sea by means of oxygen and carbon isotopes. Netherlands Journal of Sea Research, 33, 91-101.